# A system of feed-forward cerebellar circuits that extend and diversify sensory signaling

Harsh N Hariani[1,2], A Brynn Algstam[2,3], Christian T Candler[1,2], Isabelle F Witteveen[2], Jasmeen K Sidhu[2], Timothy S Balmer[2]*

[1]Interdisciplinary Graduate Program in Neuroscience, Arizona State University, Tempe, United States; [2]School of Life Sciences, Arizona State University, Tempe, United States; [3]Barrett Honors College, Arizona State University, Tempe, United States

**Abstract** Sensory signals are processed by the cerebellum to coordinate movements. Numerous cerebellar functions are thought to require the maintenance of a sensory representation that extends beyond the input signal. Granule cells receive sensory input, but they do not prolong the signal and are thus unlikely to maintain a sensory representation for much longer than the inputs themselves. Unipolar brush cells (UBCs) are excitatory interneurons that project to granule cells and transform sensory input into prolonged increases or decreases in firing, depending on their ON or OFF UBC subtype. Further extension and diversification of the input signal could be produced by UBCs that project to one another, but whether this circuitry exists is unclear. Here we test whether UBCs innervate one another and explore how these small networks of UBCs could transform spiking patterns. We characterized two transgenic mouse lines electrophysiologically and immunohistochemically to confirm that they label ON and OFF UBC subtypes and crossed them together, revealing that ON and OFF UBCs innervate one another. A Brainbow reporter was used to label UBCs of the same ON or OFF subtype with different fluorescent proteins, which showed that UBCs innervate their own subtypes as well. Computational models predict that these feed-forward networks of UBCs extend the length of bursts or pauses and introduce delays—transformations that may be necessary for cerebellar functions from modulation of eye movements to adaptive learning across time scales.

*For correspondence:
Timothy.Balmer@asu.edu

Competing interest: The authors declare that no competing interests exist.

## eLife assessment

This study presents **important** findings about synaptic connectivity among subsets of unipolar brush cells (UBCs), a specialized interneuron primarily located in the vestibular lobules of the cerebellar cortex. The evidence supporting the claims is interesting and **solid**. The work will be of interest to cerebellar neuroscientists as well as those focussed on synaptic properties and mechanisms. Although several **compelling** pieces of data were presented, some in vivo work remains to be conducted in order to test whether the hypothesis and predictions translate into the behaving animal and how it would impact the processing of feedback or feed-forward activity that would be required to promote behavior.

## Introduction

Head movement signals are encoded by the firing rate of vestibular afferents that project to the cerebellum and brainstem. The vestibular nuclei maintain a sensory representation of the head movement for tens of seconds after the vestibular signal ends through a mechanism called velocity storage

(*Fernandez and Goldberg, 1971*; *Raphan et al., 1979*). The cerebellum dynamically modulates the activity of the cells that encode velocity storage to control vestibular responses such as compensatory eye movements (*Yakushin et al., 2017*). It is unclear how the cerebellar circuit maintains a representation of a head movement long after the firing of the afferents cease.

Granule cells receive direct input from vestibular afferents, but cannot maintain the signal long enough to underlie velocity storage, because they produce only transient firing responses. The only other excitatory neuron type in the cerebellar cortex that could provide feed-forward excitation to prolong the signal to granule cells is the unipolar brush cell (UBC). UBCs are powerful circuit elements that amplify their input signals by extending them in time and projecting them to numerous granule cells (*Berthié and Axelrad, 1994*; *Mugnaini et al., 1994*; *Mugnaini et al., 2011*; *Rossi et al., 1995*; *Nunzi et al., 2001*). UBCs receive direct input from vestibular afferents, produce long-duration spiking responses, and could therefore contribute to the maintenance of a sensory representation necessary to modulate low-frequency vestibular responses (*Rossi et al., 1995*; *Mugnaini et al., 2011*; *Balmer and Trussell, 2019*). Although a single UBC can prolong its input for seconds, multiple UBCs connected in series could extend the signal much further. Indeed, there is some evidence suggesting that UBCs provide direct synaptic input to other UBCs (*Diño et al., 2000*; *Nunzi and Mugnaini, 2000*; *van Dorp and De Zeeuw, 2015*).

In addition to prolonging signals, UBCs may play important roles in diversifying granule cell firing patterns, which is essential for expansion recoding models of cerebellar learning (*Marr, 1969*; *Albus, 1971*; *Ito, 1982*; *Kennedy et al., 2014*). This diversity in UBC responses is due to different subtypes of UBCs that can be classified by their expression of specific proteins or neurotransmitter receptors (*Nunzi et al., 2002*; *Nunzi et al., 2003*; *Kim et al., 2012*; *Sekerková et al., 2014*) or by their electrophysiological response to synaptic stimulation: ON UBCs that express mGluR1 are excited by glutamate and OFF UBCs that express calretinin are inhibited by glutamate (*Borges-Merjane and Trussell, 2015*). Whether UBCs target UBCs of their own subtype, the other, or both, remains unclear and is necessary to understand how sensory signals are extended and diversified in the granule cell layer.

The connectivity of different subtypes of UBCs is difficult to address. Paired electrophysiological recordings of synaptically connected UBCs are challenging because of their long axons and their relative scarcity among the granule cells. Labeling UBCs of the same subtype with a single fluorophore provides ambiguous results—the axon terminals of UBCs look similar to UBC dendritic brushes, so a UBC axon contacting a dendritic brush of the same color would appear essentially identical to a large brush. We overcame this challenge by utilizing a Brainbow approach and combining mouse lines to label UBC subtypes with different fluorescent proteins. We find that all UBC connectivity patterns are present and develop computational models to simulate how these feed-forward circuits dynamically transform their inputs. Interconnected UBCs prolong and diversify input spiking patterns by producing an extended burst (ON to ON), extended pause (ON to OFF), pause after a delay (OFF to ON), or burst after a delay (OFF to OFF). Thus, UBCs connected in series increase the computational power of the circuit and could underlie mechanisms of cerebellar function that require transmission delays or prolonged activity.

## Results

### Electrophysiological responses of GRP and P079 UBCs reveal that they are ON and OFF UBC subtypes

To label a population of UBCs that are excited by glutamate (ON UBCs), we used a mouse line that expressed Cre recombinase (Cre) under the control of the *Grp* (gastrin-releasing peptide [GRP]) promoter (*Gerfen et al., 2013*), a promoter that had been reported to be active in mGluR1-expressing (mGluR1(+)) UBCs in a similar mouse line (*Kim et al., 2012*). We crossed the GRP-Cre mouse to a tdTomato reporter (Ai9) (*Madisen et al., 2010*) and targeted fluorescently labeled UBCs for whole-cell recording in acute brain slices (*Figure 1A*). The white matter of lobe X was stimulated with an extracellular electrode to evoke synaptic currents.

Synaptic stimulation evoked fast excitatory postsynaptic currents (EPSCs) and a slow inward current that began at the end of stimulation at higher frequencies, which is diagnostic of ON UBCs (n = 15) (*Figure 1B*). The latency between presynaptic stimulation and the beginning of the fast EPSC was 1.24 ± 0.29 ms (mean ± SD; n = 13), had a low jitter (defined as the SD of latency; 0.074 ± 0.046 ms;

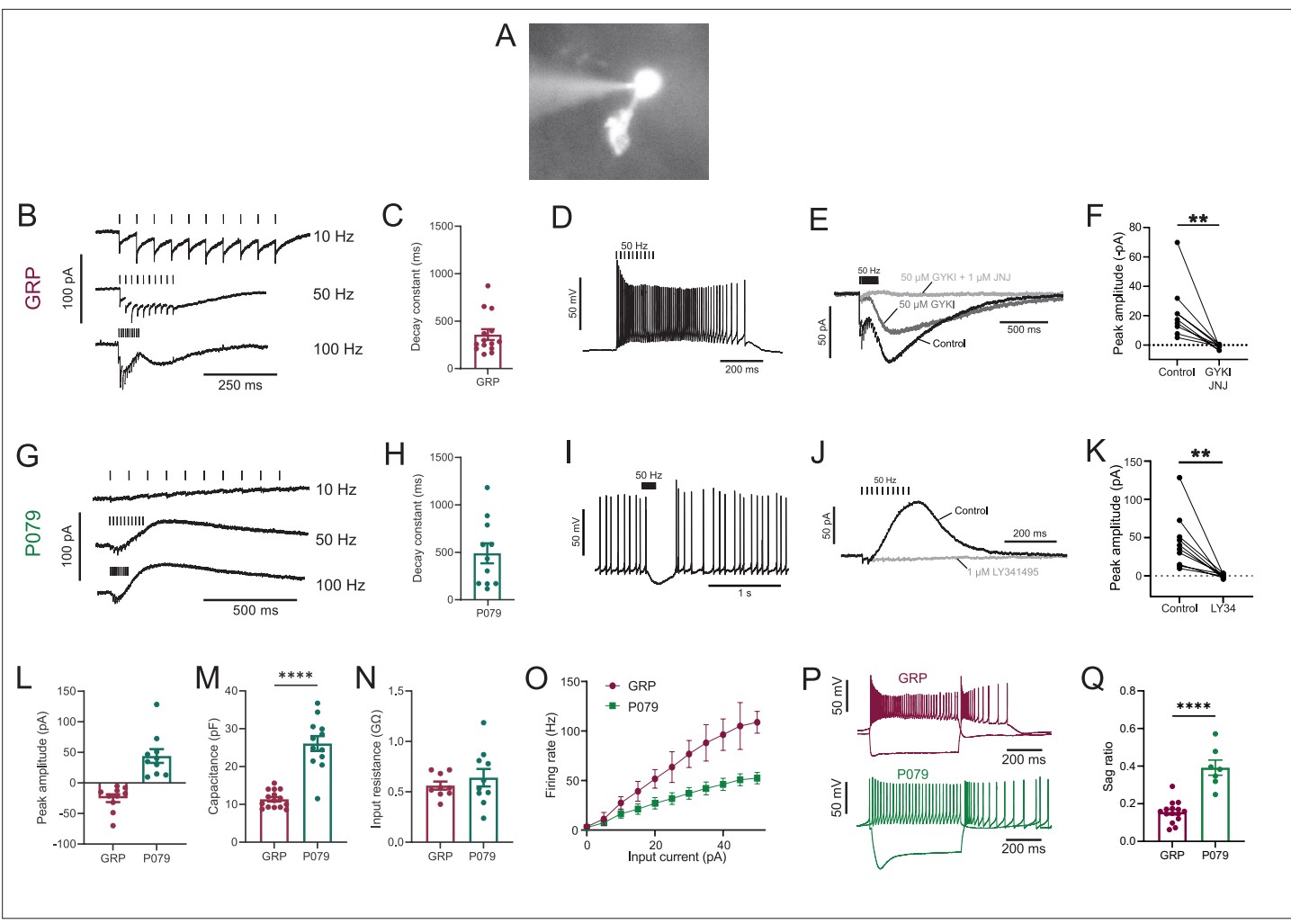

**Figure 1.** Physiological characterization of GRP and P079 unipolar brush cells (UBCs) in acute brain slices. (**A**) Representative image of UBC in whole-cell patch-clamp recording filled with Alexa Fluor 488. (**B**) In GRP UBCs, mossy fiber stimulation evoked fast excitatory postsynaptic currents (EPSCs) and a slow inward current that became more prominent with faster stimulation rates. (**C**) Summary of the decay constants of the slow EPSCs generated by a train of 10 stimuli at a 50 Hz rate in GRP UBCs. (**D**) In current clamp, 50 Hz stimulation produced a burst of spikes that outlasted the stimulus. (**E**) The fast EPSCs and part of the slow EPSC were blocked by an AMPA receptor antagonist (50 μM GYKI53655). The remaining slow EPSC was blocked by an mGluR1 antagonist (1 μM JNJ16259685). (**F**) Inward currents in GRP UBCs were consistently blocked by AMPA receptor and mGluR1 receptor antagonists. (**G**) In P079 UBCs, mossy fiber stimulation evoked slow outward currents. (**H**) Summary of the decay constants of the slow outward currents generated by a train of 10 stimuli at a 50 Hz rate in P079 UBCs. (**I**) In current clamp, 50 Hz stimulation generated a pause in spontaneous action potential firing in P079 UBCs. (**J**) The outward current was blocked by an mGluR2 antagonist (1 μM LY341495). (**K**) Outward currents in P079 UBCs were entirely blocked by 1 μM LY341495. (**L**) Peak amplitudes of the slow currents were inward in all recorded GRP UBCs and outward in all recorded P079 UBCs. (**M**) The capacitance was significantly higher in P079 UBCs compared to GRP UBCs. (**N**) The input resistance was not different between GRP UBCs and P079 UBCs. (**O**) Frequency-intensity curves show that GRP UBCs are more excitable and are able to fire at faster rates than P079 UBCs. (**P**) Example traces showing the response of GRP and P079 UBCs to 40 pA depolarizing and –100 pA hyperpolarizing 500 ms current steps. The GRP UBCs fire at a higher rate than P079 UBCs during a 40 pA depolarizing current step. The P079 UBCs have a more prominent voltage sag in response to the –100 pA hyperpolarizing current step than the GRP UBCs. (**Q**) The sag ratio (peak-steady state/peak) was larger in P079 UBCs than GRP UBCs. Stimulation artifacts have been removed for clarity. Error bars are SEM.

mean ± SD; n = 13), and was reliable, having no failures across all cells and trials, which confirms that they were monosynaptic. Note that these monosynaptic currents are most likely due to the stimulation of mossy fibers, but could also be due to stimulation of severed axons of UBCs. The slow EPSCs peaked 70.13 ± 33.45 ms after the end of the 200-ms-long 50 Hz stimulus train and their decay was well approximated by an exponential with a time constant of 358.31 ± 213.83 ms (mean ± SD; n = 14) (*Figure 1C*). In current-clamp recordings, synaptic stimulation produced a burst of action potentials in GRP UBCs that varied in duration from 510 to 900 ms (n = 3) (*Figure 1D*). The synaptic currents in GRP

UBCs were confirmed to be mediated by mGluR1 and AMPA receptors because they were blocked by JNJ16259685 and GYKI53655 (99.3 ± 2.4%; paired *t*-test; p=0.0068; n = 9; mean ± SEM) (*Figure 1E and F*). While the two drugs together blocked most of the inward current (*Figure 1F*), their individual contribution ranged widely in different cells (3–70% block by JNJ16259685 and 30–100% block by GYKI53655). This indicates that GRP UBCs are ON UBCs and their synaptic responses vary from being mediated mainly by mGluR1 to being entirely mediated by AMPA receptors.

Using the same approach, we tested the response to synaptic stimulation in a population of UBCs that expressed mCitrine in a mouse line (P079) that was generated in an enhancer trap forward genetic screen (*Shima et al., 2016*). In all P079 UBCs recorded that responded to presynaptic stimulation, a 50 Hz train evoked a slow outward current (n = 12). Faster rates of synaptic stimulation produced larger outward currents (*Figure 1G*). The slow outward currents peaked 52.7 ± 46.86 ms after the end of the 200-ms-long 50 Hz stimulus train and their decay was well approximated by an exponential with a time constant of 489.2 ± 350.4 ms (mean ± SD; n = 11) (*Figure 1H*). These outward currents hyperpolarized the cells and produced a pause in spontaneous spiking activity that varied from 0.5 to 3.9 s, averaging 1.72 ± 1.19 s (mean ± SD; n = 10) (*Figure 1I*). The outward current was mediated by mGluR2 receptors as it was blocked by the mGluR2 antagonist LY341495 (96.53 ± 1.79%; paired *t*-test; p=0.0033; mean ± SEM) (*Figure 1J and K*). These electrophysiological characteristics confirm that the P079 UBCs that can be visualized in acute brains slices are OFF UBCs.

The slow postsynaptic currents were inward in every GRP UBC and outward in every P079 UBC (*Figure 1L*). The capacitance of GRP UBCs was less than half that of P079 UBCs, which suggests that the membrane area of the OFF UBCs is greater (GRP UBCs: 11.34 pF ± 0.59, n = 15; P079 UBCs: 26.08 pF ± 1.96, n = 12; *t*-test; p<0.0001; mean ± SEM; *Figure 1M*). We did not observe a significant differ-ence in input resistance between the two cell types (GRP UBCs: 0.56 GΩ ± 0.04, n = 9; P079 UBCs: 0.64 GΩ ± 0.09, n = 10; *t*-test; p=0.44; mean ± SEM; *Figure 1N*). Current steps were injected into both UBC types to characterize their excitability. There was a significant interaction between the UBC type and response to current steps (mixed model ANOVA, p<0.0001, n = 17). GRP UBCs had a higher firing rate in response to the same level of injected current compared to P079 UBCs, despite their similar input resistance, suggesting that GRP ON UBCs are more excitable and can fire at faster rates (*Figure 1O and P*). Hyperpolarizing current steps produce a sag that is mediated by the h-current (Ih) in UBCs (*Kim et al., 2012*). P079 UBCs had a significantly more prominent sag than GRP UBCs, measured in response to a −100 pA current step (P079: 0.39 ± 0.04, n = 7; GRP: 0.16 ± 0.01, n = 15; *t*-test; p<0.0001; mean ± SEM) (*Figure 1P and Q*). This is consistent with previous work showing that OFF UBCs have a larger h-current than ON UBCs (*Kim et al., 2012*). In summary, electrophysiological recordings from GRP and P079 UBCs confirm that they are ON and OFF UBC subtypes, respectively.

## GRP and P079 UBCs are distinct subpopulations that differ in size and distribution

Besides the functional differences in the GFP and mCitrine-expressing UBCs outlined above, these neurons also showed significant differences in their size and distribution. Crossing the GRP-Cre/Ai9 mice with the P079 mice resulted in triple transgenic mice that expressed tdTomato and mCitrine in almost entirely separate subpopulations of UBCs—only 3 out of 944 UBCs that were labeled expressed both fluorescent proteins. GRP and P079 UBCs were present in the lobes of the cerebellar vermis in different densities (*Figure 2A*). More GRP UBCs than P079 UBCs were present in lobes VI–IX, with few of either subtype in lobes II–V. The highest density of both UBC subtypes was in lobe X, which is where we focused the following analyses (*Figure 2B*). UBCs have convoluted dendritic brushes that are similar in appearance to their axon terminals (*Figure 2C*).

GRP UBCs have been reported to have smaller somas than UBCs that express calretinin (calre-tinin(+)), a calcium-binding protein that labels the OFF UBC population (*Kim et al., 2012*; *Borges-Merjane and Trussell, 2015*). We found that the cross-sectional area of GRP UBCs was significantly smaller than that of P079 UBCs (GRP: 68.75 ± 9.69 μm², mean ± SD, n = 92; P079: 74.47 ± 12.53 μm², mean ± SD, n = 114, *t*-test, p=0.0004), which is consistent with our electrophysiological analysis of their membrane capacitance (*Figure 2D*). The distribution of GRP and P079 UBCs also supports their identity as ON and OFF UBCs as the density of mGluR1(+) ON UBCs is higher in the ventral leaflet of lobe X and the density of calretinin(+) OFF UBCs is higher in the dorsal leaflet (*Nunzi et al., 2002*; *Sekerková et al., 2014*).

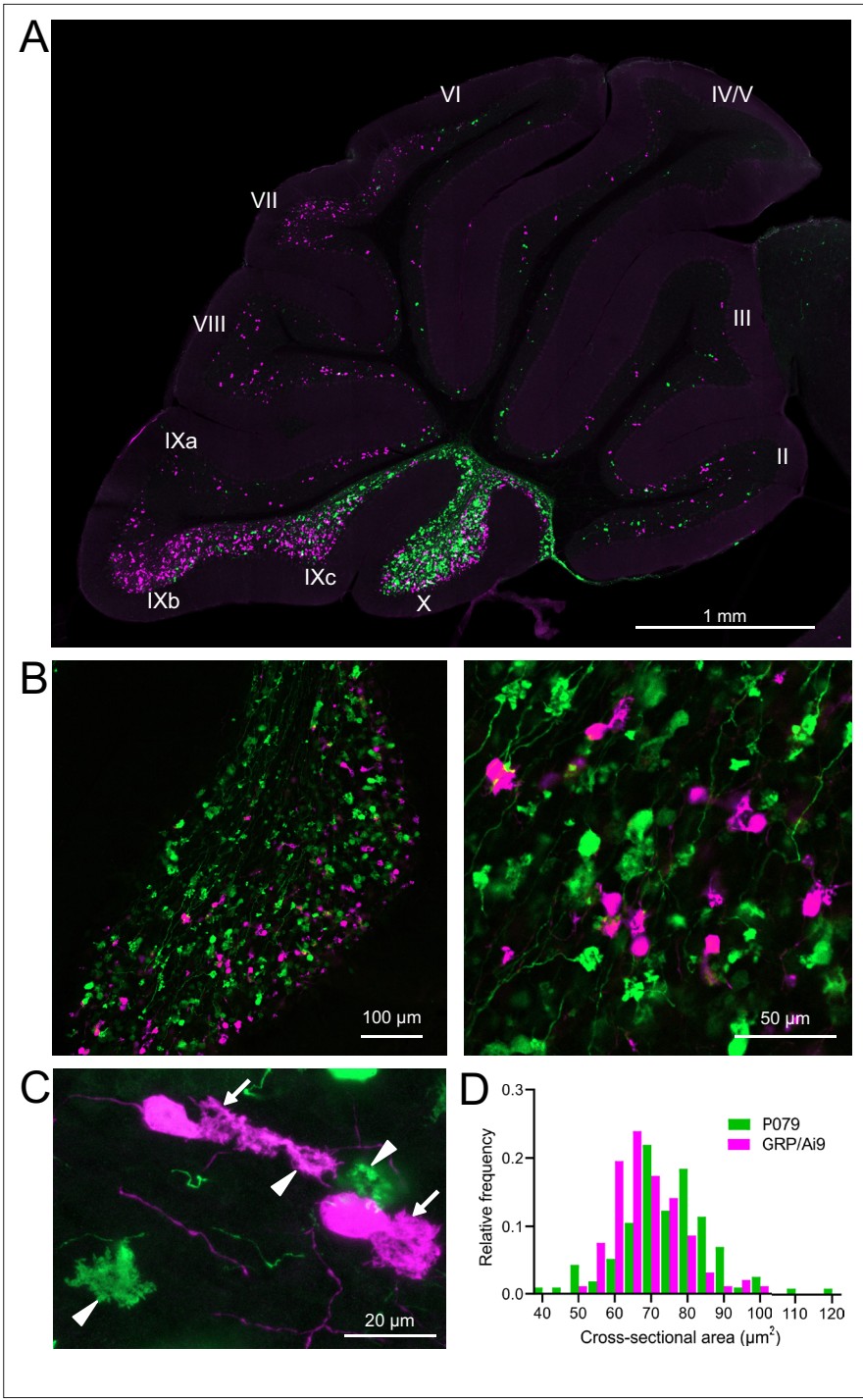

**Figure 2.** GRP/Ai9/P079 mouse line labels two distinct subsets of unipolar brush cells (UBCs). (**A**) Sagittal section of cerebellar vermis. Fluorescence is restricted to UBC cell types. P079 UBCs (mCitrine; green) are present in high density in lobe X and lobe IXc, while GRP UBCs (tdTomato, magenta) are present in those lobes as well as lobe IXb and in a lower density in lobe VI–VIII. Maximum intensity projection. (**B**) Left: sagittal section showing expression of mCitrine labeling P079 UBCs and tdTomato labeling GRP UBCs in lobe X. More GRP UBCs are present in the ventral leaflet than the dorsal leaflet. Right- UBCs and their axons and terminals are well labeled. Only rarely were UBCs labeled with both colors. Maximum intensity projection. (**C**) Magnified view of UBCs with their dendritic brushes indicated with arrows. Two P079 terminals and one GRP terminal are indicated with arrowheads. The GRP UBC at the top of the image demonstrates the challenge of differentiating the dendritic brush of a UBC from a terminal that is labeled with the same fluorophore. Maximum intensity projection. (**D**) In lobe X, P079 UBCs had larger somas than GRP UBCs, although there is a somewhat bimodal distribution in the P079 population.

## GRP UBCs are a subset of mGluR1(+) UBCs

Somatodendritic expression of mGluR1 and calretinin defines two separate populations of UBCs (termed type II and type I, respectively; *Nunzi et al., 2002*) and correspond to the functionally defined ON and OFF subtypes (*Borges-Merjane and Trussell, 2015*). To investigate the expression of mGluR1 and calretinin in GRP and P079 UBCs, we immunohistochemically labeled, imaged, and counted UBCs in both ventral and dorsal areas of lobe X in 4–8 brain slices from triple transgenic mice (n = 4 mice). We found that 98.1% (155/158) of the GRP/Ai9 UBCs expressed mGluR1 in their dendritic brushes (*Figure 3A and B*). The GRP/Ai9/mGluR1(+) population represented 20.9% (155/741) of the total population of mGluR1(+) UBCs. 0/149 GRP/Ai9 UBCs expressed calretinin in a separate series of slices (*Figure 3C and D*). Thus, the GRP line labels about 1/5th of the ON UBCs and does not label OFF UBCs.

## P079 mouse line drives mCitrine expression in the majority of calretinin(+) UBCs

Using the same triple transgenic mice and immunohistochemical approach described above, 92.2% (284/308) of the P079 UBCs were calretinin(+) (*Figure 3C and D*). The P079 UBCs represent 70.5% (284/403) of the total calretinin(+) population. We were surprised to find, however, that 61.3% (200/326) of the P079 UBCs appeared mGluR1(+). This represents 27.0% (200/741) of the total mGluR1(+) population. These P079/mGluR1(+) UBCs could be differentiated from P079/mGluR1(-) UBCs by their lower expression of mCitrine: the relative labeling intensity of the mGluR1(+) P079 UBCs, after amplification with an anti-GFP antibody, was less than half of the more common mGluR1(-) P079 UBCs (43.2% soma pixel intensity relative to mGluR1(-) P079 UBCs, n = 34 z-planes with at least one of each subtype, *t*-test, p=8.403e-6). In addition, the cross-sectional area of the mGluR1(+) P079 somas was 60.62 ± 13.67 µm$^2$, mean ± SD, n = 52 and significantly smaller than the mGluR1(-) P079 somas (72.48 ± 16.90 µm$^2$ mean ± SD, n = 40, *t*-test, p=0.00035). It is likely that the P079 UBCs that express a low density of mCitrine were not able to be visualized in our electrophysiological experiments because their native fluorescence was below the level of detection. Unfortunately, the observation that some mGluR1(+) UBCs are labeled in the P079 mouse line precludes its use for expressing genes in OFF UBCs specifically.

## GRP and P079 UBCs are distinct populations in dorsal cochlear nucleus

UBCs are also present in the dorsal cochlear nucleus, a cerebellum-like circuit in the auditory brainstem of mammals. Like granule cells in the dorsal cochlear nucleus, UBCs receive multisensory signals from various sources (*Ryugo et al., 2003*; *Balmer and Trussell, 2021b*; *Balmer and Trussell, 2022*). In order to test whether our findings about the identity of UBCs labeled in GRP-cre and P079 mice are generalizable across UBC populations, we examined their distribution in the dorsal cochlear nucleus. We found that 97.8% (132/135) of the GRP/Ai9 UBCs expressed mGluR1 in their dendritic brushes (*Figure 4A and B*). The GRP/Ai9/mGluR1(+) population represented 34.2% (132/386) of the population of mGluR1(+) UBCs and is therefore a subpopulation of about 1/3 of the mGluR1(+) ON UBCs. In a separate series of slices, 0/115 GRP/Ai9 UBCs expressed calretinin and is therefore useful for expressing genes in a subset of ON UBCs without expression in OFF UBCs in the dorsal cochlear nucleus.

We also found that 89.7% (140/156) of the P079 UBCs were calretinin(+) (*Figure 4C and D*). The P079 UBCs represent 85.9% (140/163) of the total calretinin(+) population. Similar to our results in the cerebellum, we found that 56.7% (127/224) of the P079 UBCs were mGluR1(+). This represents 32.9% (127/386) of the mGluR1(+) population. As in the cerebellum, these P079/mGluR1(+) UBCs were identifiable by their significantly lower intensity of mCitrine after antibody amplification (43.5% intensity compared to mGluR1(-) P079 UBCs, n = 19 z-planes with at least one of each type, *t*-test, p=1.352e-6). In summary, in the dorsal cochlear nucleus, the P079 mouse line expresses mCitrine in 86% of the calretinin(+) UBCs and 33% of the mGluR1(+) UBCs and the GRP line expresses Cre in 34% of the mGluR1(+) UBC population and 0% of the calretinin(+) population. Thus, the expression pattern in the dorsal cochlear nucleus of UBCs in these two mouse lines mirrors that of the cerebellum.

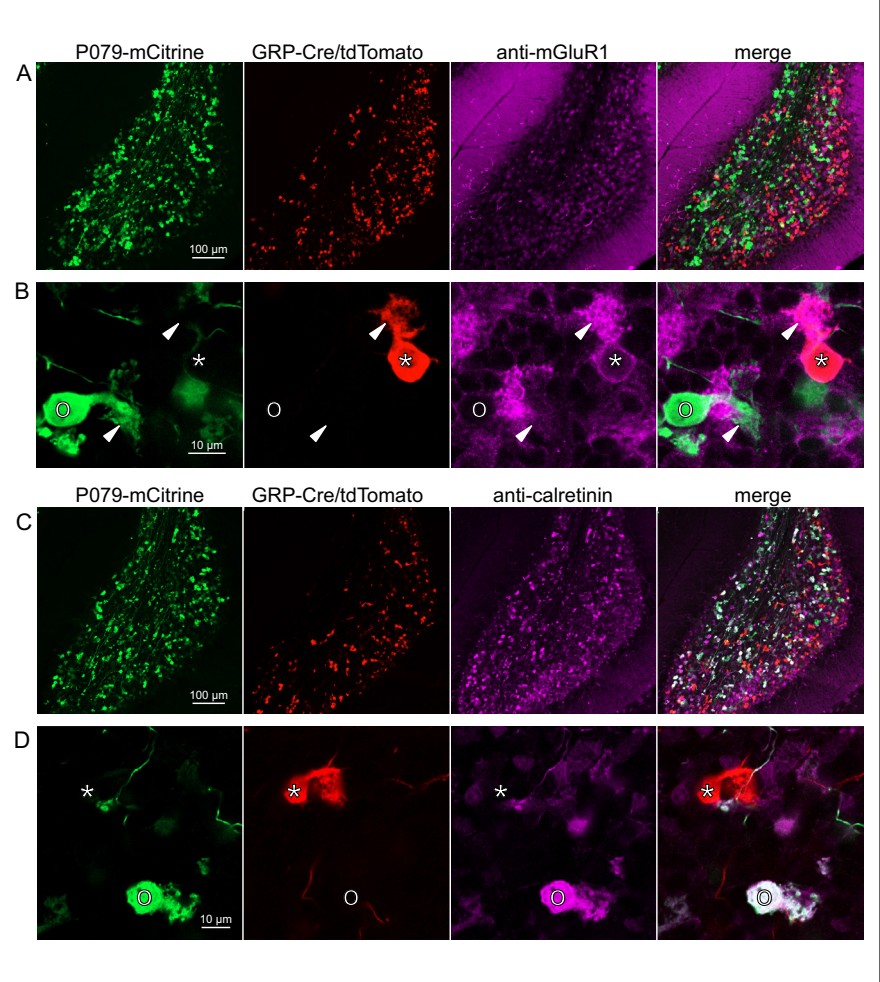

**Figure 3.** Expression of mGluR1 and calretinin in GRP and P079 unipolar brush cells (UBCs) in the cerebellum. (**A**) Sagittal section of lobe X showing genetically expressed mCitrine (green) from the P079 mouse, tdTomato (red) from the GRP-Cre/Ai9 mouse, and immunohistochemical labeling of mGluR1 (magenta). Maximum intensity projections. (**B**) Example of a P079 UBC (soma indicated with O, brush indicated with arrowhead), GRP UBC (soma indicated with *, brush indicated with arrowhead). The GRP UBCs expresses mGluR1 in the somatic membrane and dendritic brush. Most P079 UBCs do not express mGluR1. Single image planes. (**C**) Sagittal section of lobe X showing genetically expressed mCitrine (green) from the P079 mouse, tdTomato (red) from the GRP-Cre/Ai9 mouse, and immunohistochemical labeling of calretinin (magenta). Maximum intensity projections. (**D**) Example of a P079 UBC (soma indicated with O), GRP UBC (soma indicated with *). The P079 UBCs express calretinin in their cytoplasm. GRP UBCs do not express calretinin.

## Axonal projections between ON UBCs and OFF UBCs

Do ON and OFF UBC subtypes target one another? In the GRP/Ai9/P079 mouse line, all GRP UBCs are ON UBCs and the P079 UBCs are mostly OFF UBCs, although their OFF UBC identity must be confirmed by their expression of calretinin. Thus, we searched for red GRP UBC axon terminals that contacted green P079 UBC brushes that also expressed calretinin. These contacts were presumed to be synapses based on our previous anatomical analyses of functional synapses between mossy fibers and UBCs (*Balmer and Trussell, 2019*). Several examples of such synaptic connections were identified, indicating that GRP ON UBCs provide synaptic input to P079 OFF UBCs (*Figure 5A and B*). Using the same approach, we found OFF UBC axon terminals (P079 and calretinin(+)) contacting ON UBC brushes (GRP) (*Figure 5C and D*). These projections were rare and we did not attempt to characterize the proportion of ON to OFF UBC connections as both populations are subsets of ON and OFF UBCs and would be a significant underestimate. However, these results do reveal that UBC subtypes synapse on one another and may introduce complex spiking patterns.

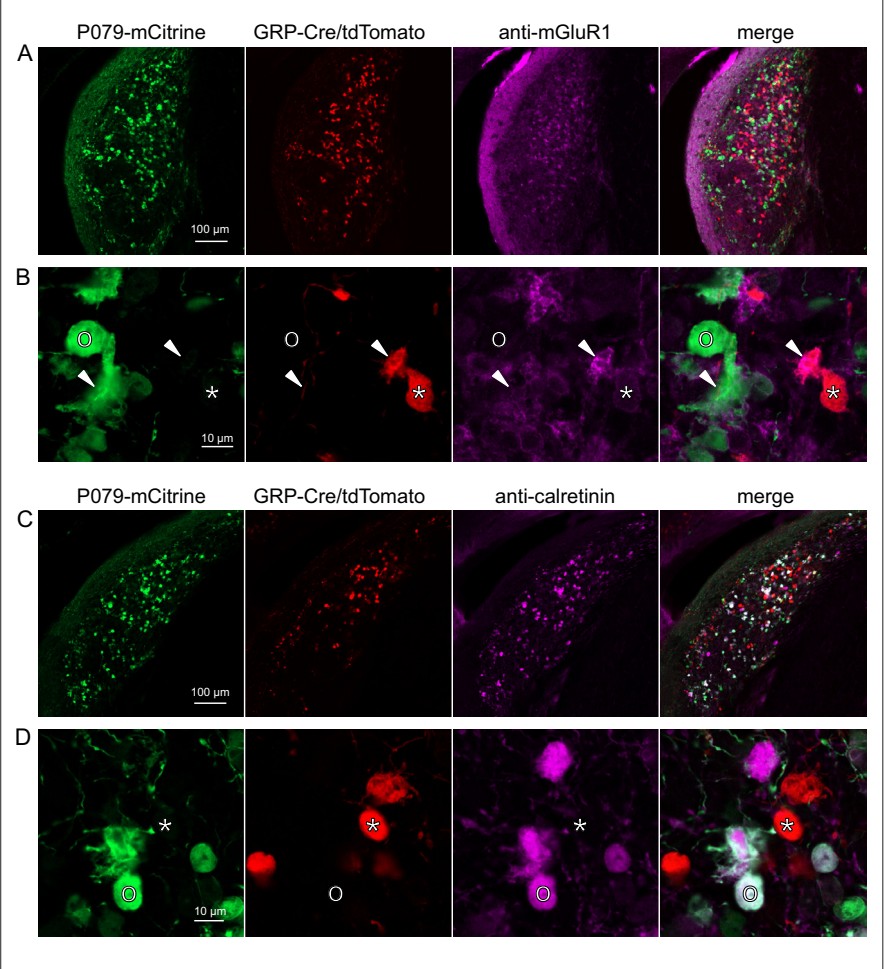

**Figure 4.** Expression of mGluR1 and calretinin in P079 and GRP unipolar brush cells (UBCs) in the dorsal cochlear nucleus. (**A**) Coronal section of the dorsal cochlear nucleus showing genetically expressed mCitrine (green) from the P079 allele, tdTomato (red) from the GRP-Cre/Ai9 alleles, and immunohistochemical labeling of mGluR1 (magenta). Maximum intensity projections. (**B**) Example of a P079 UBC (soma indicated with O, brush indicated with arrowhead), GRP UBC (soma indicated with *, brush indicated with arrowhead). The GRP UBCs expresses mGluR1 in the somatic membrane and dendritic brush. Most P079 UBCs do not express mGluR1, although some that are weakly labeled do appear to express mGluR1. (**C**) Coronal section of the dorsal cochlear nucleus showing genetically expressed mCitrine (green) from the P079 allele, tdTomato (red) from the GRP-Cre/Ai9 alleles, and immunohistochemical labeling of calretinin (magenta). Maximum intensity projections. (**D**) Example of a P079 UBC (soma indicated with O), GRP UBC (soma indicated with *). The P079 UBCs express calretinin in their cytoplasm. GRP UBCs do not express calretinin.

## Axonal projections from ON UBCs to other ON UBCs

To test whether ON UBCs synapse on other ON UBCs, we searched for labeled axon terminals in the GRP/Ai9 mouse that contacted brushes of ON UBCs labeled with mGluR1. Two examples are shown in *Figure 6*. Care was taken to confirm that mGluR1 labeling that appeared to decorate the brush of a UBC could be traced to a soma with the characteristic expression in the somatic membrane that forms a circle. These examples show that GRP ON UBCs target mGluR1(+) UBCs that are not transgenicially labeled in the GRP mouse line.

In a second approach to investigate synaptic connections between ON UBCs, we utilized a Brainbow2.1-Confetti reporter mouse line to label GRP ON UBCs with distinct colors (*Livet et al., 2007*; *Snippert et al., 2010*). In the offspring resulting from the GRP-Cre/Brainbow2.1-Confetti cross, expression of the fluorescent proteins was low and it was therefore necessary to enhance with anti-GFP and anti-mCherry antibodies. In doing so, both membrane-bound mCerulean and cytoplasmic YFP

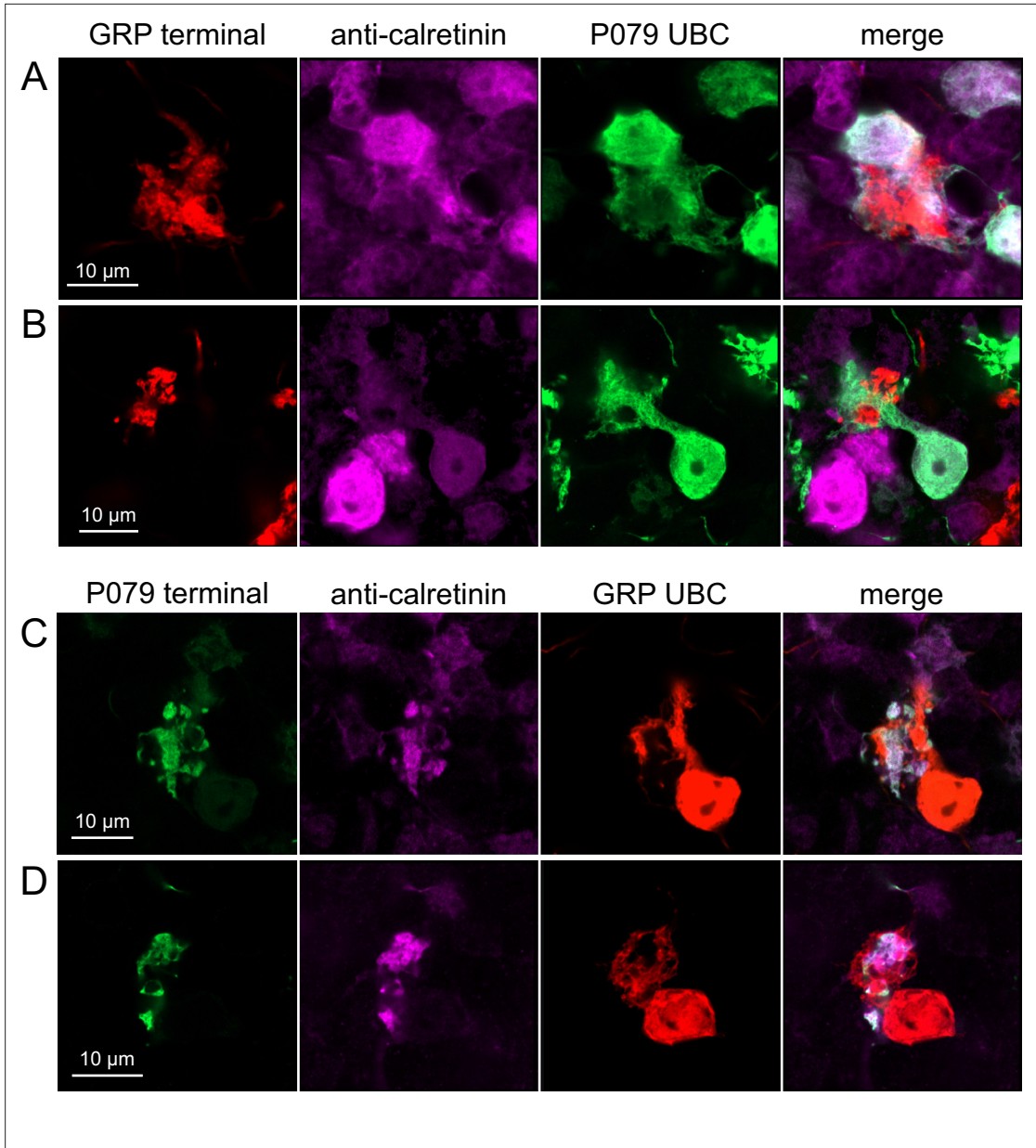

**Figure 5.** Anatomical evidence showing axonal projections between ON and OFF unipolar brush cell (UBC) subtype. (**A, B**) ON UBC axon terminals (GRP, red) that project to OFF UBCs (P079, green), confirmed to express calretinin (magenta). Single image sections. (**C, D**) OFF UBC axon terminals (P079, green), confirmed to express calretinin (magenta) contacting the dendritic brushes of ON UBCs (GRP, red). Single image sections.

were labeled with the anti-GFP antibody, and cytoplasmic tdimer2(12) was labeled with anti-mCherry. This resulted in various combinations of expression that could be identified by wavelength (green or red) and cellular localization (membrane-bound or cytoplasmic). In most labeled UBCs, multiple alleles were expressed and could be visualized in the same cells, perhaps owing to the sensitivity of the antibody amplification (*Figure 7A*). Several examples of GRP UBC axon terminals contacting the brushes of other GRP UBCs were identified (*Figure 7B–D*). Thus, GRP ON UBCs project to other GRP ON UBCs in a feed-forward excitatory circuit that is likely to prolong the duration of an excitatory signal.

## Axonal projections from OFF UBCs to other OFF UBCs

To address whether OFF UBCs project axons to other OFF UBCs, we exploited the observation that the P079 line expresses mCitrine in some, but not all, calretinin-expressing OFF UBCs. Thus, when

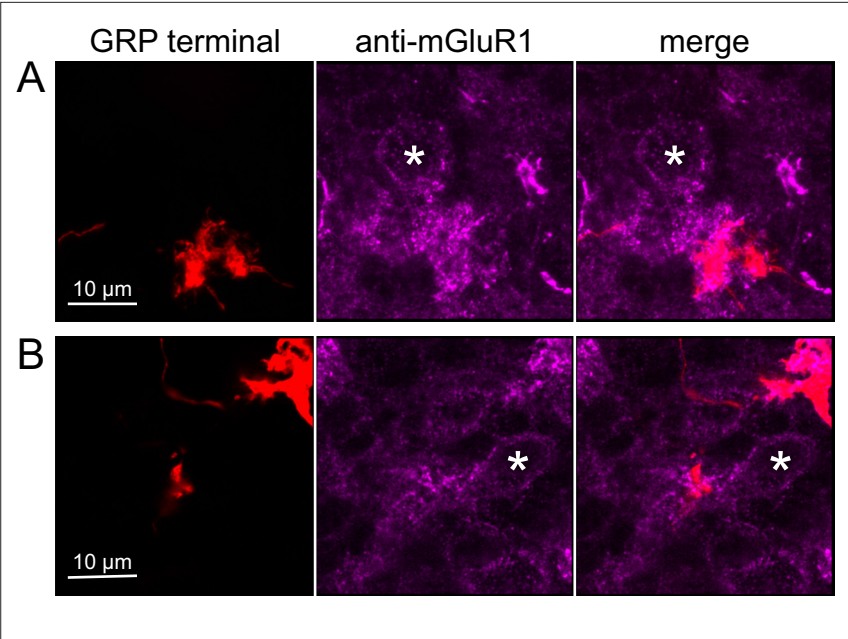

**Figure 6.** Anatomical evidence for synaptic connections between ON unipolar brush cells (UBCs). (**A, B**) GRP ON UBC terminals (red) contacting mGluR1(+) ON UBCs (magenta). Somas indicated with *. Single image sections.

calretinin is labeled immunohistochemically using a non-green fluorophore, all OFF UBCs are revealed and their brushes can be examined for the presence of connections from mCitrine-labeled P079 OFF UBCs axon terminals. Several examples for OFF to OFF UBC synaptic connections were confirmed anatomically (*Figure 8*). In some cases, calretinin(+) axon terminals innervated the brushes of P079 UBCs that were also calretinin(+) (*Figure 8A and B*). In other cases, the P079 UBC terminal contacted calretinin(+) UBCs that were not labeled transgenically in the P079 line (*Figure 8C*).

Brainbow2.1-Confetti reporter mice were crossed with a line that expresses Cre in calretinin(+) neurons after induction by tamoxifen (*Taniguchi et al., 2011*). Tamoxifen induced the expression of fluorescent proteins in a subset of these calretinin(+) OFF UBCs and were amplified with antibodies as above (*Figure 9A*). Several examples are shown, confirming that calretinin(+) OFF UBCs provide input to one another (*Figure 9B–D*). In one of these examples, an axon terminal appeared to contact the soma of a postsynaptic UBC (*Figure 9C*), which has been previously reported to occur between extrinsic mossy fibers and UBC somata (*Balmer and Trussell, 2019*). In another case, an axon terminal appeared to project a narrow fiber to the edge of the dendritic brush of a postsynaptic UBC (*Figure 9D*). The presence of large or small contacts implies differences in synaptic strength, suggesting that OFF UBCs may provide strong synaptic signals to one another (*Figure 9B*) or presumably weaker signals (*Figure 9C and D*) that are subsequently integrated with other inputs.

## Computational models predict transformations of spiking patterns through synaptically connected ON and OFF UBCs

To address how spiking activity may be transformed by different types of UBCs, computational models were developed to simulate the essential features of ON and OFF UBCs, including their passive electrical properties, synaptic current responses, and spiking patterns. UBCs can fire spontaneously or be silent in acute brain slices (*van Dorp and De Zeeuw, 2015*; *Kim et al., 2012*). We chose to make the OFF UBCs spontaneously active in this model so that synaptic input would produce a pause in firing (OFF response) that could be easily measured. By contrast, we made ON UBCs not spontaneously active so that synaptic input would produce a burst of action potentials (ON response). Action potentials triggered glutamate concentration transients that drove the conductance of model AMPA receptors in ON UBCs and mGluR2 in OFF UBCs (see 'Materials and methods'). We contrasted simple cases of an excitatory input to either an ON or OFF UBC, with a more complex case in which the excitatory

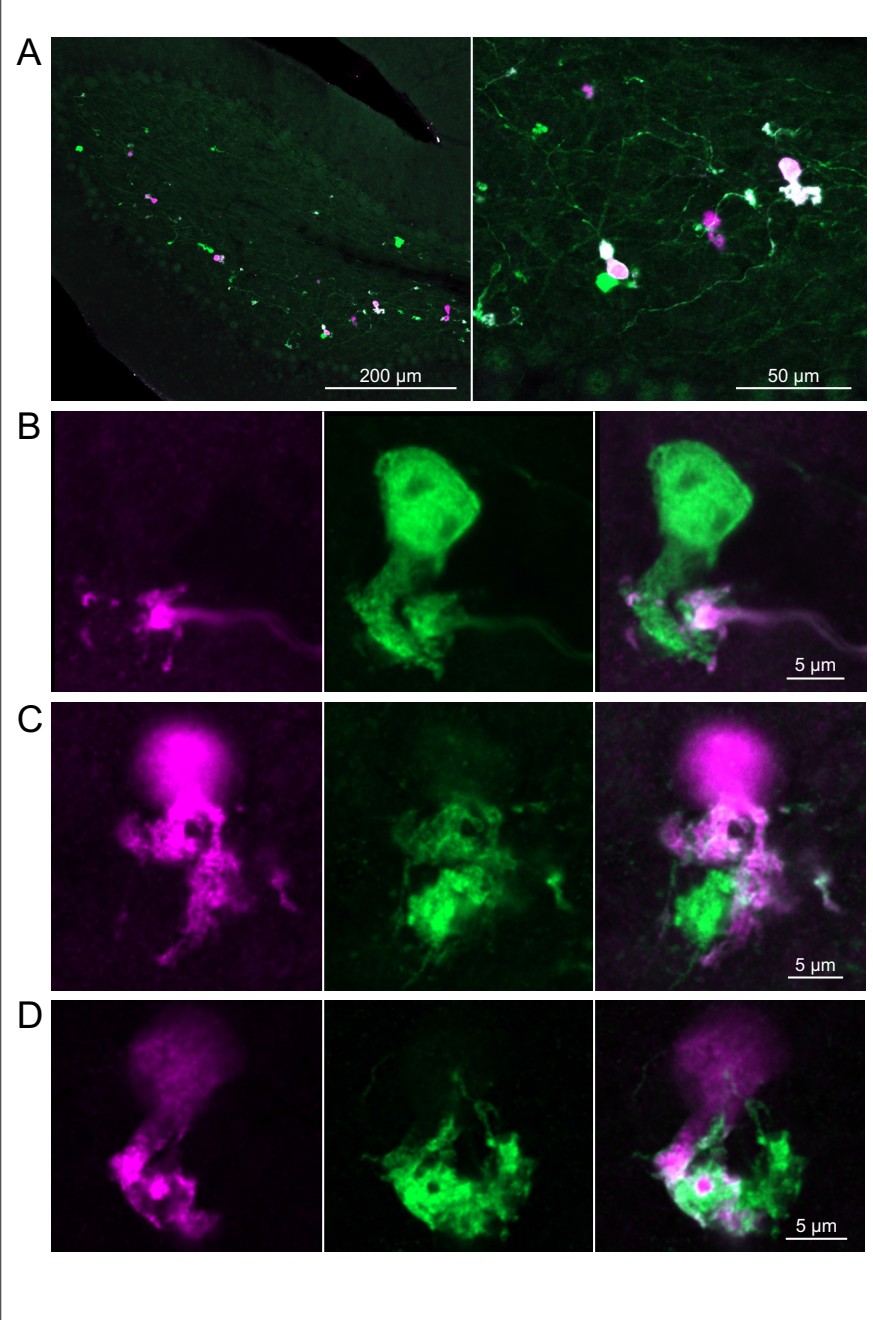

**Figure 7.** Brainbow reporter labels synaptic connections between GRP ON unipolar brush cells (UBCs). (**A**) GRP ON UBCs expressed fluorescent proteins that were either amplified with the anti-GFP antibody or anti-mCherry antibody. GRP UBCs and their axons and terminals are labeled. Maximum intensity projections. (**B**) Example of a GRP UBC expressing cytoplasmic YFP (green) that appears to be contacted by a presynaptic terminal from the axon of another GRP UBC that expresses both cytoplasmic YFP (green) and cytoplasmic tdimer2(12) (magenta). Single image sections. (**C**) Example of GRP ON UBC expressing cytoplasmic tdimer2(12) (magenta) and membrane-bound mCerulean (green) that appears to receive a synaptic terminal that expresses cytoplasmic YFP (green). Single image sections. (**D**) Example of GRP ON UBC that expresses cytoplasmic tdimer2(12) (magenta) receiving input from a GRP UBC axon that expresses cytoplasmic YFP (green). Single image sections.

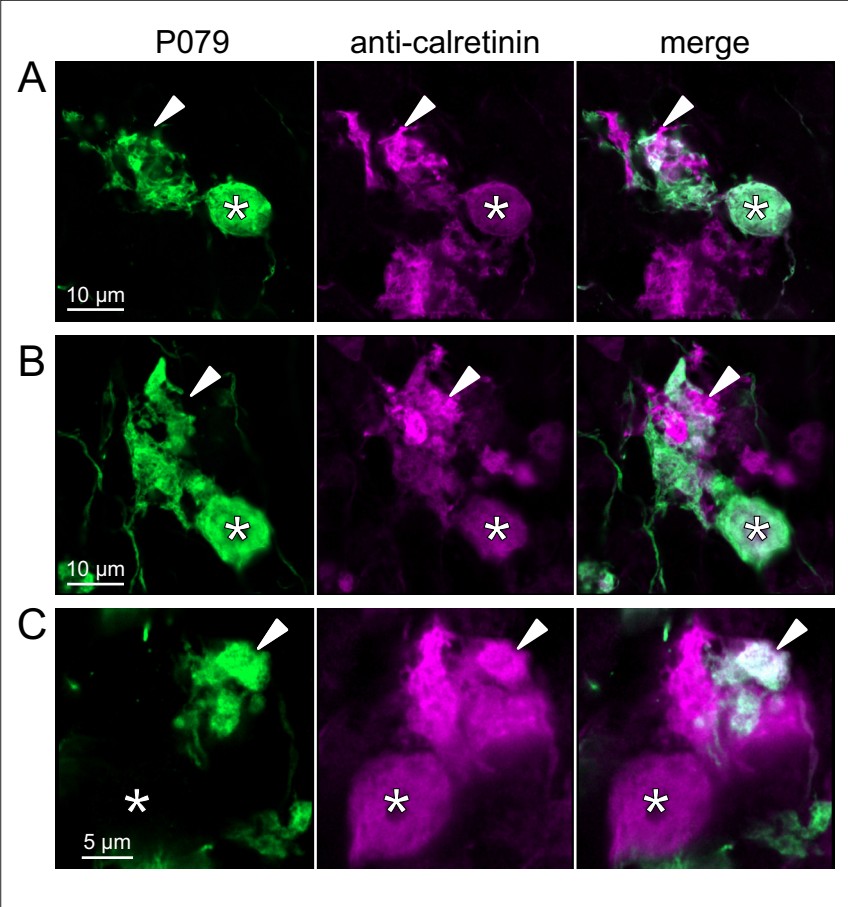

**Figure 8.** Anatomical evidence for synaptic connections between OFF unipolar brush cells (UBCs). (**A, B**) Examples of P079 OFF UBCs (green) that are confirmed to express calretinin (magenta) and are contacted by calretinin(+) terminals that are not labeled in the P079 line. Single image sections. (**C**) Example of an OFF UBC that expresses calretinin (magenta) that is contacted by a P079 (green) axon terminal that expresses calretinin (magenta). Somas are indicated with *, terminals are indicated with arrowheads. Single image sections.

input drove an 'intermediate' UBC, which then synapsed on a second UBC. In this way, we could examine the predicted outcomes of ON to ON, ON to OFF, OFF to ON, and OFF to OFF scenarios.

## Serial ON UBCs multiply burst duration

Presynaptic input to an ON UBC produced bursts of action potentials that multiplied in duration in a circuit with an intermediate ON UBC (*Figure 10A–C*). Three presynaptic spikes in a burst lasting 19.6 ms produced a burst of 12 action potentials in the first UBC, lasting 358.0 ms, and 35 action potentials in the second UBC, lasting 1175.3 ms, ~60 times longer than the initial presynaptic input (*Figure 10A and B*). The enhancement of the spiking duration both in the first and second ON UBC was more pronounced with shorter bursts of presynaptic input and began to plateau at longer durations of input (*Figure 10C*). The plateau in spiking duration occurs because the rebound AMPA receptor-mediated slow EPSC that occurs at the end of synaptic stimulation in ON UBCs reaches a maximum after ~100 ms of synaptic stimulation (*Lu et al., 2017*).

## Intermediate ON UBCs extend the pause in postsynaptic OFF UBCs

A burst of presynaptic spikes causes a pause in spontaneous firing in OFF UBCs due to the outward current produced by mGluR2 (*Figure 10D*). When an intermediate ON UBC was present between the presynaptic axon and the OFF UBC, the synaptic signal to the OFF UBC was extended, and therefore increased the mGluR2 current, markedly lengthening the duration of the pause in spiking (*Figure 10E and F*). Indeed, the presence of an intermediate ON UBC increased the duration of the pause in the

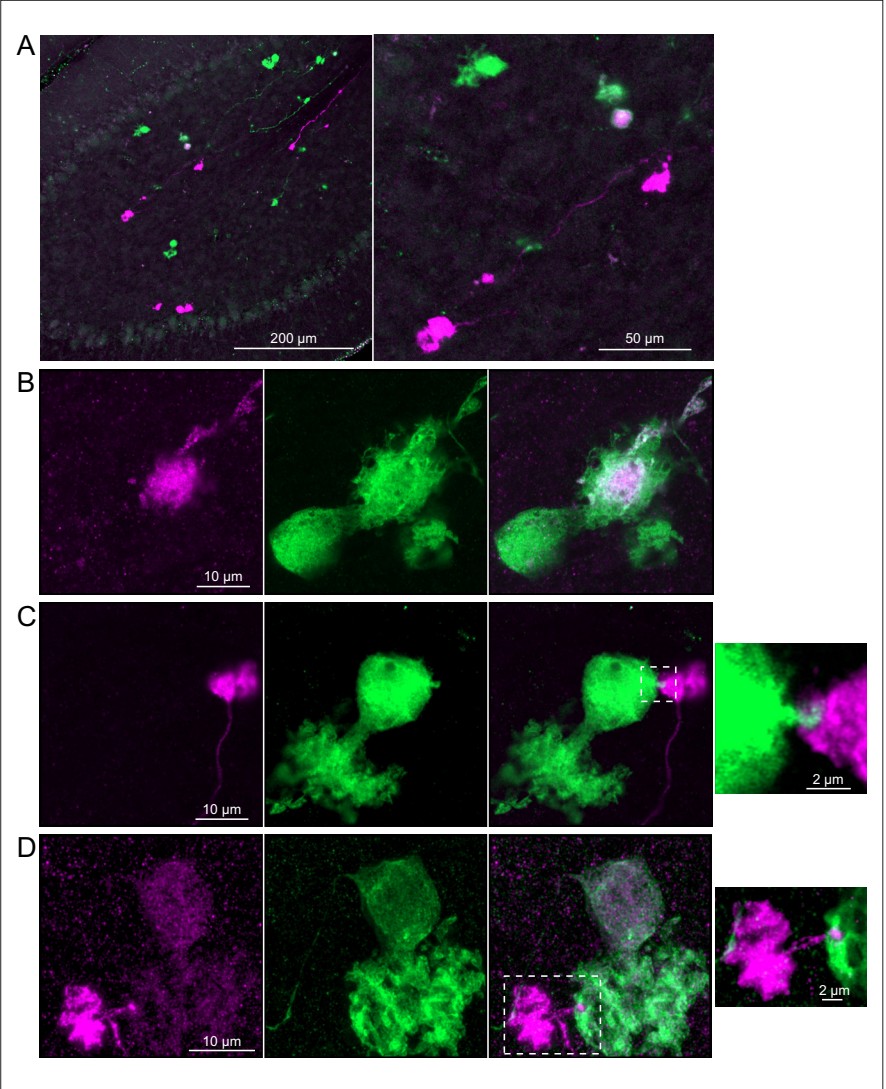

**Figure 9.** Brainbow reporter labels synaptic connections between OFF unipolar brush cells (UBCs) in calretinin-Cre mouse line. (**A**) Calretinin-Cre UBCs expressed fluorescent proteins that were either amplified with the anti-GFP antibody or anti-mCherry antibody, revealing UBCs and their axons and terminals in lobe X. Maximum intensity projections. (**B**) Example of a labeled calretinin-Cre axon terminal (magenta) that contacted the brush of a calretinin-Cre UBC (green), showing that these OFF UBC project to one another. (**C**) Example of a calretinin-Cre UBC axon terminal (magenta) that appeared to contact a spine-like extension from the soma of another calretinin-Cre UBC (green). (**D**) Example of a calretinin-Cre axon terminal (magenta) that projected a small fiber that made a bouton-like synapse onto the brush of another calretinin-Cre UBC (green) that is presumed to have another unlabeled input to the majority of its dendritic brush.

postsynaptic OFF UBC by ~940 ms following 1–10 presynaptic action potentials (indicated by an upward shift in the curve in *Figure 10G*). The small difference in how much the pause was lengthened across the range of 1–10 presynaptic spikes is due to the intermediate ON UBC's response—even a single presynaptic spike produces a burst of action potentials that is only slightly extended by longer trains of presynaptic input. Across a population of ON UBCs, however, the size and duration of the slow inward currents that drive their firing vary and would thus extend the pause of a postsynaptic OFF UBC for different durations.

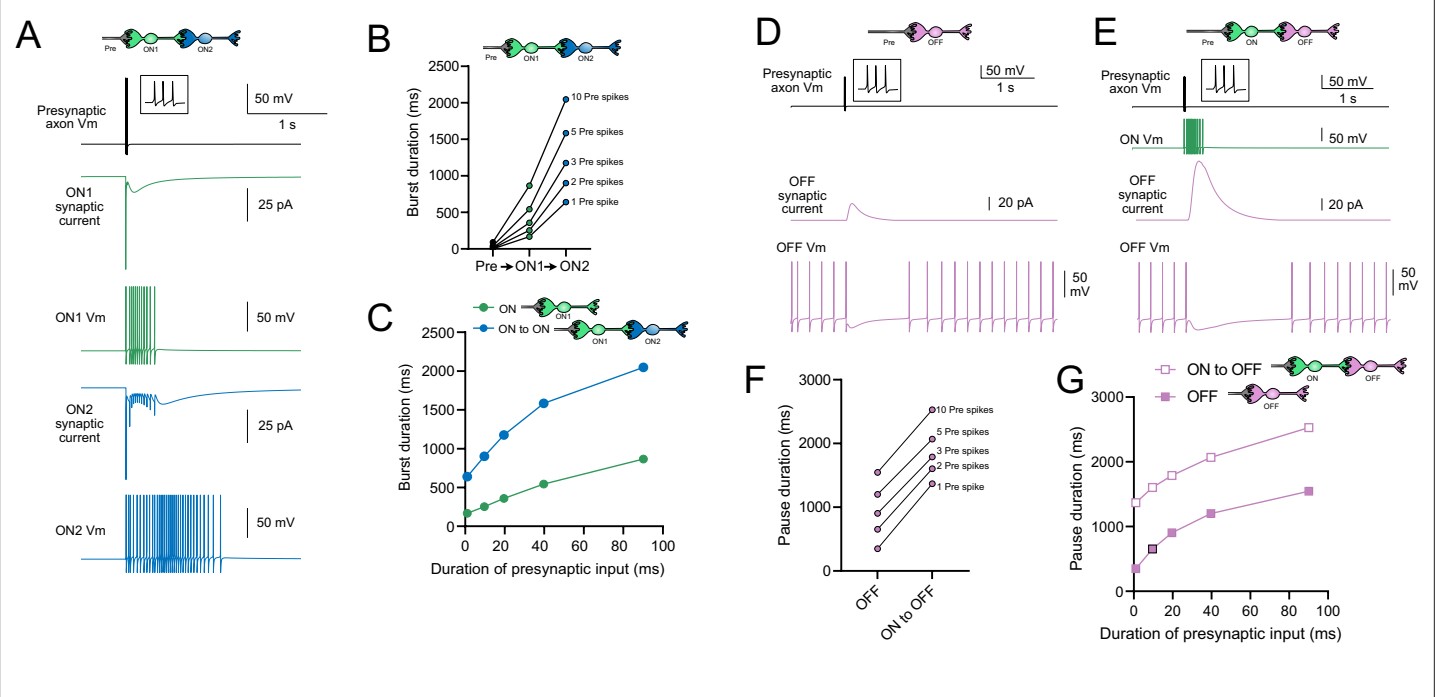

**Figure 10.** Model predicts that intermediate ON unipolar brush cells (UBCs) extend signal. (**A**) Model in which a presynaptic axon fired three action potentials (inset shows presynaptic spikes with expanded time scale) that generated an AMPA receptor-mediated synaptic current and a burst in spikes in the first ON UBC (ON1), which produced an extended AMPA receptor-mediated synaptic current in the postsynaptic UBC (ON2) and an extended burst of spikes. (**B**) Presynaptic spikes were amplified to longer bursts in each subsequent ON UBC. (**C**) Burst duration in the intermediate ON UBC (green) and second ON UBC (blue) as a function of presynaptic input duration. (**D**) Model in which a presynaptic axon fired three action potentials that generated an mGluR2-mediated synaptic current in an OFF UBC, which produced a pause in spontaneous firing lasting ~1 s. (**E**) An intermediate ON UBC between the presynaptic axon and OFF UBC generated a larger and longer-lasting mGluR2-mediated current that produced an extended pause in spontaneous firing. (**F**) Pause duration in an OFF UBC was extended by an intermediate ON UBC. (**G**) Pause duration in an OFF UBC with and without an intermediate ON UBC as a function of presynaptic input duration.

## Intermediate OFF UBCs produce a delayed pause in postsynaptic ON UBC

Because OFF UBCs fired spontaneously in our model, an ON UBC postsynaptic to an intermediate OFF UBC fired with an irregular spiking pattern (*Figure 11A*). Presynaptic input to the intermediate OFF UBC paused both its firing and its release of glutamate onto the postsynaptic ON UBC, which therefore produced a pause in the postsynaptic ON UBC. However, this pause only occurred after a delay of about 370 ms due to the slow AMPA receptor-mediated EPSC in the ON UBC that decayed slowly until it no longer drove spiking. The length of the delay before the pause began is related to the strength of the AMPA receptor-mediated current in the postsynaptic ON UBC. The length of the pause in the ON UBC depends on how long the intermediate OFF UBC pauses. Thus, an intermediate OFF UBC converts the usual excitatory ON UBC response to a pause that is similar to that of the OFF UBC itself. However, the pause is distinct from that of OFF UBCs because it is delayed and briefer (*Figure 11B*).

## Serial OFF UBCs produce a delayed burst in postsynaptic OFF UBC

Finally, we considered the case of a connection between two OFF UBCs. In a circuit with a spontaneously firing OFF UBC synapsing onto another OFF UBC (labeled OFF1 and OFF2, respectively, in *Figure 11C and D*), the tonic glutamatergic input and mGluR2 inhibition prevented spontaneous spiking in the postsynaptic cell. When the intermediate OFF UBC received a burst of presynaptic input, it was inhibited by its own mGluR2 current and therefore stopped firing, thereby disinhibiting the postsynaptic OFF UBC, which then fired after a delay (*Figure 11C*). Note that this firing would be more pronounced had we incorporated calcium and TRP conductances that contribute to a 'late-onset

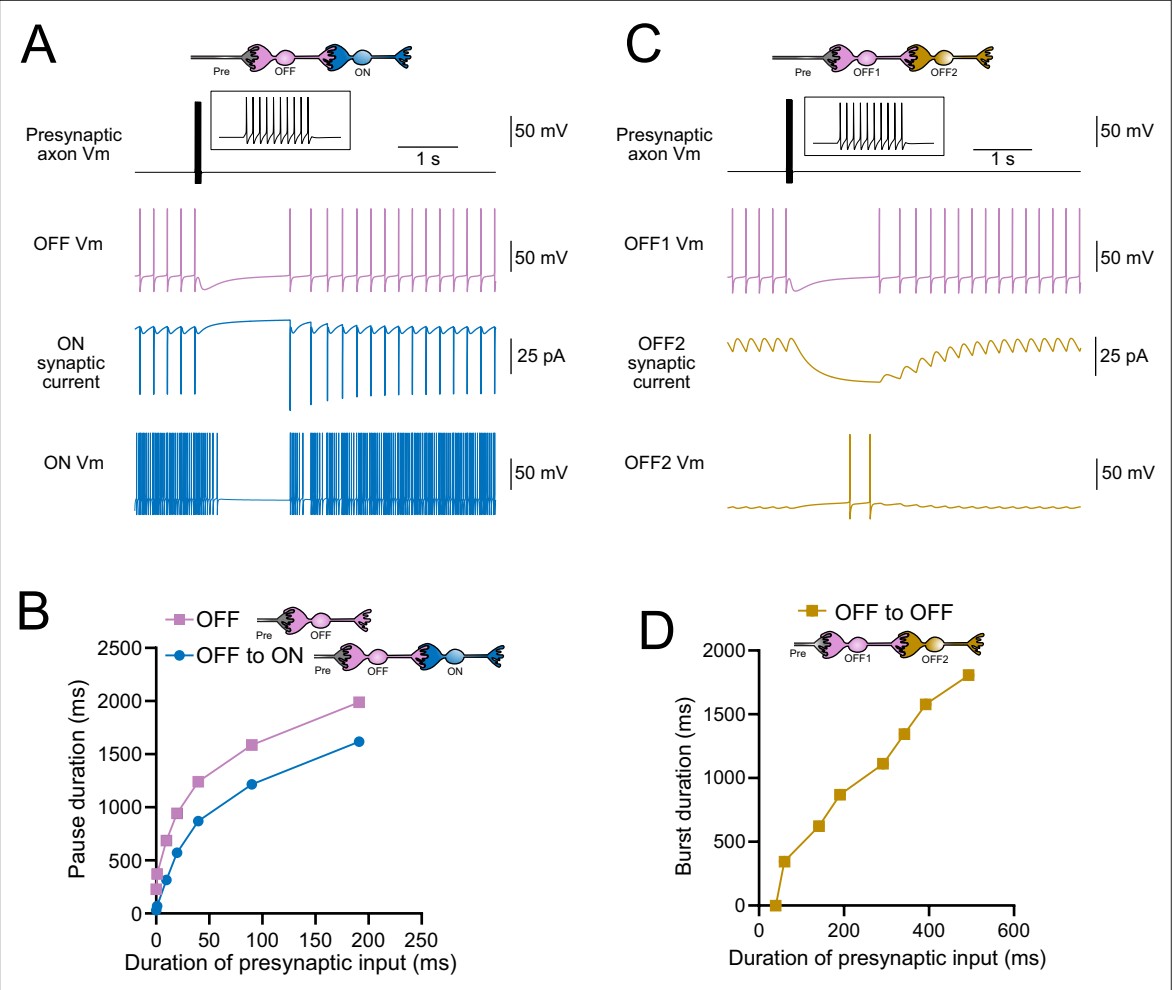

**Figure 11.** Model predicts that intermediate OFF unipolar brush cells (UBCs) delay signals. (**A**) Model showing an intermediate OFF UBC (magenta) that produced a delayed pause in a postsynaptic ON UBC (blue). The spontaneous activity of the intermediate OFF UBC drove irregular spiking in the postsynaptic ON UBC. Presynaptic input (10 spikes shown in inset with expanded time scale) produced a pause in the intermediate OFF UBC, which stopped its release of glutamate onto the ON UBC, and produced a pause after a delay that is due to the slow decay of the AMPA receptor-mediated current. (**B**) The intermediate OFF UBC (magenta) paused for longer than the postsynaptic ON UBC (blue) because the pause in the ON UBC occurred after a delay and it ended as soon as the intermediate OFF UBC resumed firing. (**C**) Model showing an intermediate OFF UBC that produced delayed spikes in a postsynaptic OFF UBC. Spontaneous firing of the intermediate OFF UBC (OFF1, magenta) tonically inhibited the postsynaptic OFF UBC (OFF2, gold). Presynaptic input caused a pause in the intermediate OFF UBC, which disinhibited the postsynaptic OFF UBC, allowing it to fire with a delay that depended on the decay of the mGluR2 current. (**D**) The duration of the burst of action potentials in the postsynaptic UBC increased with longer durations of presynaptic input.

response' in UBCs (*Locatelli et al., 2013*; *Subramaniyam et al., 2014*). The duration of the delayed firing in the second postsynaptic OFF UBC depended on the duration of presynaptic input signal to the intermediate UBC (*Figure 11D*). Moreover, the delay between the last spike in the intermediate OFF UBC and the first spike in the postsynaptic OFF UBC was contingent on the slow decay of the mGluR2 current and was 1082.7 ms in the example in *Figure 11C*.

In summary, our modeling suggests that feed-forward circuits of ON and OFF UBCs transform presynaptic signals by producing an extended spike burst (ON to ON), extended pause (ON to OFF), pause after a delay (OFF to ON), or burst after a delay (OFF to OFF) (*Figure 12*). This diversification of spiking patterns and durations may be important for maintaining sensory representations of movement and mechanisms of adaptive learning.

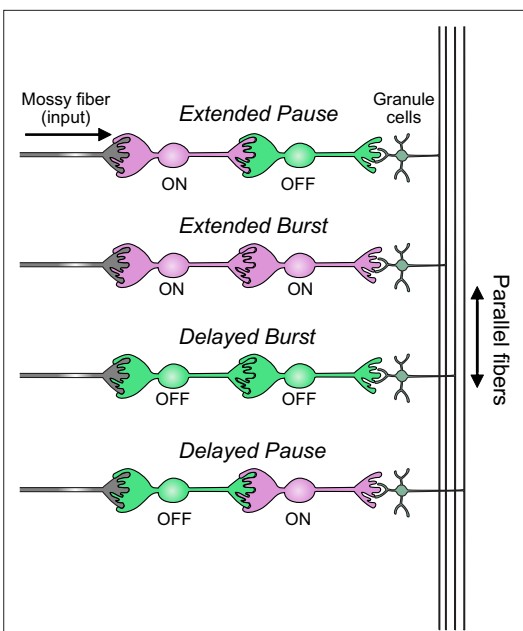

**Figure 12.** Summary of transformations of spiking patterns in unipolar brush cell (UBC) circuits. Four patterns of connectivity between UBC subtypes were anatomically defined and computationally modeled. Intermediate ON UBCs extended the usual pause or burst in firing of the postsynaptic OFF or ON UBC. Intermediate OFF UBCs delayed and inverted the response of the postsynaptic UBC, producing either a delayed burst in OFF UBCs or a delayed pause in ON UBCs.

## Discussion
### GRP and P079 mouse lines label distinct UBC subtypes that form parallel and convergent pathways

Transgenic mice were generated that expressed a green fluorescent protein in OFF UBCs and a red fluorescent protein in ON UBCs in the same tissue. Virtually no neurons expressed both fluorescent proteins. Immunohistochemical analysis showed that all GRP UBCs expressed mGluR1 and most P079 UBCs expressed calretinin in both cerebellum and dorsal cochlear nucleus. GRP and P079 UBCs had different average soma sizes and distributions in the cerebellum. Whole-cell recordings in acute brain slices confirmed that the GRP UBCs produce excitatory ON responses mediated by mGluR1 and AMPA receptors, and that P079 UBCs produce inhibitory OFF responses mediated by mGluR2. Their membrane capacitance and excitability were so distinct that the distributions of these properties were almost nonoverlapping. These results provide further evidence that some UBC subtypes are genetically and functionally distinct, despite the observation that when the entire population of UBCs is examined, their electrophysiological responses form a continuum from entirely excited by glutamate to entirely inhibited by glutamate (*Guo et al., 2021*; *Kozareva et al., 2021*).

The UBCs in these mouse lines expressed the fluorescent proteins throughout their axons and presynaptic terminals, which revealed for the first time that axon terminals from one UBC subtype contact the dendric brushes of the other UBC subtype. We also showed that UBCs of the same subtype provide input to one another by labeling GRP or calretinin-Cre UBCs with different fluorescent proteins using the Brainbow2.1-Confetti reporter line. This result was confirmed with additional approaches that utilized the incomplete labeling of mGluR1(+) ON UBCs in the GRP line and the incomplete labeling of calretinin(+) OFF UBCs in the P079 line. Earlier work reported that mGluR1(+) UBCs and calretinin(+) UBCs form parallel pathways (*Nunzi et al., 2002*). Our current data extends the previous result by showing that the even more specific subtypes of GRP or P079 UBCs provide input to members of their own subtype. These networks of UBCs could provide the granule layer with a circuit mechanism to diversify and extend the pattern of spiking produced by mossy fiber input.

Anatomically identifiable connections between UBCs were not present in all brain slices and finding them required a careful search. UBC labeling was sparse due to the highly specific genetic labeling techniques and further sparsification by the Brainbow reporter, which made it impossible to accurately estimate the density of these UBC to UBC connections. Electrophysiological evidence suggests that UBC to UBC connections are not common because spontaneous EPSCs that would indicate a spontaneously firing presynaptic UBC are not frequently observed in intracellular recordings of UBCs in acute brain slices. In an analysis of feed-forward excitation of granule layer neurons, only 4 out of 140 UBCs were reported to have this indirect evidence of a firing presynaptic UBC (*van Dorp and De Zeeuw, 2015*), which suggests that UBC to UBC connections may be rare. On the other hand, previous work using organotypic slice cultures from P8 mice estimated that 2/3 of the UBC population receives input from other UBCs (*Nunzi and Mugnaini, 2000*). This suggests a much higher density of UBC to UBC connections, but could be due to the young age of the brains used, which is before UBCs

have matured (*Morin et al., 2001*), and also due to increased collateral sprouting that can occur in culture (*Jaeger et al., 1988*). Another study imaged 2–4-week-old rat cerebellar slices at an electron microscopic level and found that 4 out of 14 UBC axon terminals contacted UBC brushes (*Diño et al., 2000*). Future work is necessary to accurately estimate the density and impact of feed-forward UBC circuits.

The purpose of this study was to reveal the postsynaptic targets of UBCs with a focus on ON and OFF UBC subtypes. However, it is important to emphasize that the final output and main target of UBCs are granule cells. Most of the labeled UBC axon terminals did not synapse on UBCs and are therefore presumed to synapse on numerous granule cells. Even the axon terminals that targeted UBCs likely also contacted granule cells. Thus, UBCs are interneurons that, when positioned between mossy fibers and granule cells, modify the input to granule cells and have the capacity to synchronize the firing of large ensembles of granule cells via their branching axons and large terminals.

The approaches used here were not able to determine the existence of networks of more than two UBCs connected one after the other. If present, three or more UBCs in series could extend and transform the input in even more dramatic ways. The temporal diversity that UBC circuits generate may underlie the flexibility of the cerebellum to coordinate movements over a broad range of behaviors.

## Limitations of the model

Here we addressed how feed-forward glutamatergic excitation and inhibition is transformed from one UBC to the next depending on their subtype. The model focuses on AMPA receptor-mediated excitation and mGluR2-mediated inhibition. One limitation of the model is that it does not consider feed-forward and lateral inhibition from Golgi cells, which shapes the spiking of UBCs in response to afferent stimulation. Golgi cells receive mossy fiber input and inhibit UBCs through their corelease of GABA and glycine (*Dugué et al., 2005*; *Rousseau et al., 2012*). Golgi cells control the temporal dynamics of the firing of granule cells as well as their gain (*Rossi et al., 2003*; *Kanichay and Silver, 2008*) and are critical to larger-scale dynamics of the cerebellar cortical network (*D'Angelo, 2008*). Purkinje cells provide additional inhibition to ON UBCs that could influence how UBC circuits transform signals (*Guo et al., 2016*). A more complex model that implements Golgi cells and other critical circuit elements will be needed to investigate the role of feed-forward UBC circuits in cerebellar network dynamics and motor behaviors in vivo.

## UBC circuits generate a broad range of firing patterns and may be essential for cerebellar learning

Vestibular afferents that encode head movements synapse on both UBCs and granule cells. Granule cells that receive this input are likely to fire in phase with the input signal. A single intermediate UBC would shift the phase of firing of its postsynaptic granule cells, depending on the UBC's response to glutamatergic input, which varies from nearly complete inhibition (e.g., P079 OFF UBC) to complete excitation (e.g., GRP ON UBC) (*Guo et al., 2021*; *Kozareva et al., 2021*). For example, an intermediate ON UBC between a presynaptic input and another UBC extended the signal markedly in our models. In circuits with an intermediate OFF UBC, the postsynaptic UBCs signal was inverted: ON UBCs paused when they were no longer driven by glutamate, and OFF UBCs fired when they were no longer inhibited by glutamate. The signal that granule cells subsequently receive would be a burst or pause in firing that lasts for hundreds or thousands of milliseconds. We suggest that these transformations could provide a circuit mechanism for maintaining a sensory representation of movement for seconds, which is thought to be necessary to modulate low-frequency vestibular responses such as compensatory eye movements.

The delays introduced by an intermediate UBC were due to the slow decay of the excitatory or inhibitory synaptic currents produced by the presynaptic input. Other cellular and synaptic properties that would shape the timing of pauses, delays, and bursts include the amount of glutamate released and the rate of its removal, the geometry of the extracellular space, glutamate receptor number and localization, short-term synaptic plasticity, currents underlying rebound spiking, and intrinsic excitability (*Kim et al., 2012*; *Kinney et al., 2013*; *Locatelli et al., 2013*; *Zampini et al., 2016*; *Lu et al., 2017*; *Balmer et al., 2021a*). The combination of numerous mechanisms that shape the signal as it is transmitted along the granule layer circuit could produce tunable delay lines essential for sensory representation on an enormous range of time scales.

Extending the duration of a sensory feedback signal may also be essential for motor learning when there is a significant delay between the motor output and the sensory feedback, for example, when throwing an object at distant target, a challenge referred to as the credit assignment problem (*Sutton and Barto, 1981*; *Raymond and Lisberger, 1998*; *Suvrathan, 2019*). There are a variety of ways that the cerebellum could extend a signal in time, such as slow axonal conduction, long membrane time constants, inhibition followed by rebound spiking, and slow intracellular processes (*Braitenberg, 1967*; *Rossi et al., 1995*; *Fiala et al., 1996*; *Hooper et al., 2002*; *D'Angelo and De Zeeuw, 2009*; *van Dorp and De Zeeuw, 2014*). UBCs circuits represent another mechanism that could prolong granule cell firing and contribute to the maintenance of sensory signals that underlie motor learning in behaviors with delayed sensory feedback.

The diversification of spiking patterns by intermediate UBCs may contribute to the pattern separation that is necessary for theories of cerebellar learning (*Marr, 1969*; *Albus, 1971*; *Ito, 1982*; *Kennedy et al., 2014*; *Zampini et al., 2016*). In one of the best understood cerebellum-like circuits, the electrosensory lobe of electric fish, UBCs are essential to produce temporally diverse and delayed spiking patterns that are necessary for accurate cancellation of self-generated signals (*Kennedy et al., 2014*). The authors report that a post-inhibitory rebound that occurred in UBCs may account for the delayed spiking patterns. The presence of synaptic pathways with multiple UBCs could contribute to these long pauses and delays that may be critical for adaptive learning on time scales ranging from hundreds of milliseconds to tens of seconds.

## Sources of synaptic input to UBC subtypes

The firing rates of putative UBCs recorded in vivo in the vestibular cerebellum of mammals follow head velocity and eye movements, in some cases with delays of hundreds of milliseconds (*Simpson et al., 2005*; *Barmack and Yakhnitsa, 2008*; *Hensbroek et al., 2015*). Our data suggests that these long delays could be implemented by feed-forward circuits containing multiple UBCs. It is unknown whether UBCs that are labeled in the GRP or P079 mouse lines are a subtype that receives a specific source of input. In lobe X, ON UBCs defined by their electrophysiological response or mGluR1-expression receive input from both primary vestibular afferents and second-order mossy fibers from the vestibular nucleus, while OFF UBCs only receive vestibular nuclear input (*Balmer and Trussell, 2019*). Understanding whether the UBCs labeled in these mouse lines receive head or eye movement signals, integrated signals that encode head orientation relative to gravity, or another type of signal would help elucidate their specific roles in cerebellar functions.

# Materials and methods

## Animals

Mice of both sexes were used from the following mouse lines and their crosses: P079 line: Et(tTA/mCitrine)P079Sbn (*Shima et al., 2016*); GRP-Cre line: Tg(Grp-Cre)KH107Gsat (MMRRC_031182-UCD) (*Gerfen et al., 2013*); Ai9 line: Gt(ROSA)26Sor^tm9(CAG-tdTomato)Hze (IMSR_JAX:007909) (*Madisen et al., 2010*); Brainbow2.1-Confetti line: Gt(ROSA)26Sor^tm1(CAG-Brainbow2.1)Cle/J (IMSR_JAX:017492) (*Snippert et al., 2010*); and calretinin-Cre line: Calb2^tm2.1(cre/ERT2)Zjh/J (IMSR_JAX:013730) (*Taniguchi et al., 2011*). Cre-mediated recombination was induced in calretinin-Cre mice by intraperitoneal injections of 10 mg/ml tamoxifen (T5648, Sigma) in corn oil at a dose of 75 mg/kg per day for 2–3 d and perfused 2–3 wk later. Mice were bred in a colony maintained in the animal facility managed by the Department of Animal Care and Technologies, and all procedures were approved by Arizona State University's Institutional Animal Care and Use Committee under protocol #21-1817R. Transgenic mice were genotyped by light at P0–P3 or by PCR.

## Brain slice preparation

For acute brain slice electrophysiology, male and female P079 and GRP-Cre/Ai9 mice were used at ages P21–35. Animals were deeply anesthetized with isoflurane and the brain was extracted under ice-cold high-sucrose artificial cerebral spinal fluid (ACSF) containing the following (in mM): 87 NaCl, 75 sucrose, 25 NaHCO$_3$, 25 glucose, 2.5 KCl, 1.25 NaH$_2$PO$_4$, 0.4 Na-ascorbate, 2 Na-pyruvate, 0.5 CaCl$_2$, 7 MgCl$_2$, bubbled with 5% CO$_2$/95% O$_2$. Parasagittal cerebellum sections 200–300 μm thick were cut with a vibratome (7000smz-2, Campden Instruments) in ice-cold high-sucrose ACSF. Slices

**Table 1.** ON unipolar brush cell (UBC) conductances.

**ON UBC**

| Modeled conductance | gmax (S/cm2) | Erev (mV) |
| --- | --- | --- |
| $g_{Na}$ | 0.1 | 50 |
| $g_K$ | 0.03 | –90 |
| $g_H$ | 1e-5 | –30 |
| $g_{pas}$ | 2e-4 | –65 |
| $g_{K-slow}$ | 8e-4 | –90 |
| Diameter | 20 μm | |

recovered at 35°C for 30–40 min, in ACSF containing the following (in mM): 130 NaCl, 2.1 KCl, 1.2 $KH_2PO_4$, 3 Na-HEPES, 10 glucose, 20 $NaHCO_3$, 0.4 Na-ascorbate, 2 Na-pyruvate, 1.5 $CaCl_2$, 1 $MgSO_4$, bubbled with 5% $CO_2$/95% $O_2$ (300–305 mOsm). Slices were maintained at room temperature (~23°C) until recording. Recordings were performed from lobe X of the cerebellum within 6 hr of preparation.

## Electrophysiological recordings

Acute brain slices were perfused with ACSF using a peristaltic pump (Ismatec) at 2–3 ml/min and maintained at 32–34°C with an inline heater (Warner Instruments). ACSF contained 0.5 μM Strychnine and 5 μM SR95531 to block synaptic inhibition. The recording setup was composed of an Olympus BX51 fixed-stage microscope with Dodt gradient contrast optics, and 4× and 60× water immersion Olympus objectives. UBCs were initially identified by soma size or transgenic fluorescence and confirmed by intracellular filling with 5 μM Alexa Fluor 568 or 488 hydrazide sodium salt (Life Technologies). Patch electrodes were pulled with borosilicate glass capillaries (OD 1.2 mm and ID 0.68 mm, AM Systems) with a horizontal puller (P1000, Sutter Instruments). Intracellular recording solution contained (in mM) 113 K-gluconate, 9 HEPES, 4.5 $MgCl_2$, 0.1 EGTA, 14 Tris-phosphocreatine, 4 $Na_2$-ATP, 0.3 tris-GTP, 0.1–0.3% biocytin, 290 mOsm, pH 7.2–7.25. All recordings were corrected for a –10 mV junction potential. Data were acquired using a Multiclamp 700B amplifier and pClamp 11 software (Molecular Devices). Signals were acquired with 5–10× gain, sampled at 50–100 kHz using a Digidata (1550A, Molecular Devices) analog-digital converter, and low-pass filtered at 10 kHz, with further filtering applied offline. Patch pipettes tip resistance was 5–8 MΩ; series resistance was compensated with correction 20–40% and prediction 50–70%, bandwidth 2 kHz. Membrane potential was held constant at –70 mV in voltage-clamp experiments. Electrical stimulation was performed using a concentric bipolar electrode (CBBPC75, FHC) placed in the white matter of the sagittal cerebellar slice. Stimuli were evoked using a stimulus isolation unit (Iso-Flex, A.M.P.I.) delivering 100-250 μs duration pulses of 0–90 V.

**Table 2.** OFF unipolar brush cell (UBC) conductances.

**OFF UBC**

| Modeled conductance | gmax (S/cm2) | Erev (mV) |
| --- | --- | --- |
| $g_{Na}$ | 0.1 | 50 |
| $g_K$ | 0.03 | –90 |
| $g_H$ | 3.1e-4 | –30 |
| $g_{pas}$ | 7e-5 | –62 |
| $g_{K-slow}$ | 8e-4 | –90 |
| Diameter | 27 μm | |

**Table 3.** Glutamate diffusion at AMPA and mGluR2 receptors.

| Glutamate diffusion parameter | AMPA synapse | mGluR2 synapse |
|---|---|---|
| Molecules released (#) | 3e6 | 3e5 |
| Distance (nm) | 750 | 1750 |
| Diffusion coefficient (cm²/s) | 0.33 | 0.33 |
| Tortuosity | 1.55 | 1.55 |
| Volume fraction | 0.21 | 0.21 |
| Ambient glutamate (mM) | 0.005 | 0 |

## Immunohistochemistry and imaging

Mice were overdosed with isoflurane and perfused through the heart with 0.01 M phosphate-buffered saline, 7.4 pH (PBS) followed by 4% paraformaldehyde in PBS. Brains were extracted from the skull and incubated in 4% paraformaldehyde in PBS overnight at 4°C. 50-μm-thick sections were made on a vibratome (7000smz-2, Campden Instruments). To recover cells that were filled with biocytin during whole-cell recording, acute brain slices were fixed 1–2 hr in 4% paraformaldehyde in PBS, followed by storage in PBS. Both floating 50 μm sections and 200–300-μm-thick acute slices were treated with the following procedures. Sections were rinsed 3 × 5 min in PBS, blocked, and permeabilized in 5% BSA, 2% fish gelatin, 0.2% Triton X-100 in PBS for >1 hr at room temperature. Sections were incubated in primary antibodies overnight in 1% fish gelatin in PBS at 4°C on an orbital shaker. Primary antibodies included chicken polyclonal anti-GFP (1:2000, Aves Labs, GFP-1020, AB_10000240), goat polyclonal anti-mCherry (1:2000, Sicgen, AB0040, AB_2333093), mouse monoclonal anti-rat mGluR1a (1:800, BD Pharmingen, 556389, AB_396404), and rabbit polyclonal anti-calretinin (1:2000, Swant, 7697, AB_10000342). Sections were rinsed 3 × 5 min in PBS, followed by secondary antibodies (1:500) and streptavidin (1:2000) that were diluted in 1% fish gelatin in PBS and incubated overnight at 4°C on an orbital shaker. Secondary antibodies included donkey anti-chicken Alexa Fluor 488 (Jackson ImmunoResearch, 715-545-155, AB_2340375), donkey anti-mouse Alexa Fluor 647 (Jackson ImmunoResearch, 715-605-151, AB_2340863), donkey anti-rabbit Alexa Fluor 647 (Jackson ImmunoResearch, 711-605-152, AB_2492288), donkey anti-rabbit Cy3 (Jackson ImmunoResearch, 711-165-152, AB_2307443), donkey ant-goat Cy3 (Jackson ImmunoResearch, 705-165-147, AB_2307351), and streptavidin-Alexa Fluor 647 (Thermo Fisher Scientific, S21374, AB_2336066). Sections were mounted on microscope slides (Superfrost Plus, Fisher Scientific) and coverslipped with Fluoromount-G (Southern Biotech). Images were acquired using a confocal microscope (LSM800, Zeiss) with the Airyscan system that reconstructs super-resolution images from a series of images acquired under spatially structured illumination. Images are single optical planes unless otherwise specified.

## Image quantification

Sagittal brain slices containing lobe X were imaged systematically across the medial–lateral axis. One area in the dorsal region and one area in the ventral region of lobe X in each selected slice were imaged using a 63× objective. The volume was 202.8 μm × 202.8 μm × 10.8 μm for each image using ~12 z-planes. Images were annotated for transgenic expression of tdTomato or mCitrine as well as mGluR1 or calretinin labeling. tdTomato, mCitrine, and calretinin labeling is cytoplasmic, and UBCs were easily counted if they had a labeled soma and attached dendritic brush. mGluR1 is mostly expressed in the dendrite, but labeling is also present in the soma membrane, which can be observed as a circle. mGluR1 UBCs were counted if there was labeling in the dendritic brush that was attached

**Table 4.** Synaptic conductances.

| Synaptic receptor | gmax (S/cm²) | Erev (mV) |
|---|---|---|
| ON UBC, AMPA | 4e-9 | 0 |
| OFF UBC, mGluR2 | 1e-8 | –90 |

to an apparent mGluR1-labeled soma. This necessarily conservative method of identifying mGluR1-labeled UBCs likely resulted in an underestimation of the total number of mGluR1(+) UBCs.

The labeling intensity of mCitrine in P079 UBCs that were mGluR1(+) was compared to P079 UBCs that were mGluR1(-) by measuring the mean pixel intensity in the somas of each subtype of UBCs. At least one of each subtype was measured in the same image and same z-plane to account for differences in brightness that could be due to differences in immunohistochemical labeling, image acquisition settings, and depth in the slice (deeper cells are less well labeled due to more limited antibody penetration). For each z-plane with at least one mGluR1(+) and one mGluR1(-) P079 UBC, a difference in brightness was calculated and reported as a relative brightness of the mGluR1(+) P079 UBCs to the brighter mGluR1(-) P079 UBCs.

## Computational modeling

Single-compartment models of ON and OFF UBCs were built using NEURON (*Hines and Carnevale, 1997*; *Carnevale and Hines, 2006*) and utilized fast voltage-gated sodium ($g_{Na}$) and potassium ($g_K$) conductances to produce action potentials (*Destexhe et al., 1994*), a slow voltage-gated potassium conductance from a model of cerebellar granule cells ($g_{K\text{-slow}}$) (*D'Angelo et al., 2001*), and a passive leak conductance ($g_{pas}$). A hyperpolarization-activated conductance ($g_H$) (*Kim et al., 2012*; *Subramaniyam et al., 2014*) was added to ON and OFF UBCs to produce a voltage sag in response to hyperpolarizing current pulses that reproduced those reported in *Figure 1*. Input resistance of the cells was measured using a –5 pA current pulse and adjusted to approximate GRP ON UBCs and P079 OFF UBCs by adjusting the passive leak conductance. Capacitance was measured from the membrane time constant and adjusted by changing the membrane area. Specific membrane capacitance was set to 1 µF/cm². *Tables 1 and 2* show the parameters used in the ON and OFF UBC models.

Synaptic transmission was implemented by applying glutamate transients simulated with a three-dimensional diffusion equation to an AMPA receptor model that was fit to ON UBC data (*Lu et al., 2017*) and described in detail in our previous work (*Balmer et al., 2021a*; *Tables 3 and 4*). OFF UBCs have mGluR2 receptors that are thought to be distant from the presynaptic release sites (*Jaarsma et al., 1998*), which we approximated by reducing the amount of glutamate released by a factor of 10 and increasing the distance by 1 µm in the diffusion equation (*Table 3*). To model mGluR2 currents in P079 UBCs that are mediated by G-protein-coupled inwardly rectifying potassium channels, a GABA-B receptor model (*Destexhe et al., 1998*) was modified to approximate the kinetics of our recorded currents by increasing the unbinding rate from 0.02 ms⁻¹ to 0.0215 ms⁻¹. A train of 10 presynaptic release events produced an 865-ms-long burst in the ON UBC model and a 1545 ms pause in the OFF UBC model, which is within the range observed in our data set.

## Acknowledgements

Funding was provided by the NIH/NIDCD R00 DC016905, Hearing Health Foundation, and National Ataxia Foundation. We thank Dr. Jason Newbern for confocal microscope use, Dr. Sacha Nelson for the P079 mouse line, and Dr. Larry Trussell for comments on the manuscript.

## Additional information

### Funding

| Funder | Grant reference number | Author |
|---|---|---|
| National Ataxia Foundation | | Timothy S Balmer |
| Hearing Health Foundation | | Timothy S Balmer |
| National Institute on Deafness and Other Communication Disorders | DC016905 | Timothy S Balmer |

The funders had no role in study design, data collection and interpretation, or the decision to submit the work for publication.

## Author contributions
Harsh N Hariani, A Brynn Algstam, Data curation, Formal analysis, Investigation, Visualization, Writing – review and editing; Christian T Candler, Isabelle F Witteveen, Jasmeen K Sidhu, Data curation, Formal analysis, Investigation, Writing – review and editing; Timothy S Balmer, Conceptualization, Resources, Data curation, Software, Formal analysis, Supervision, Funding acquisition, Validation, Investigation, Visualization, Methodology, Writing – original draft, Project administration, Writing – review and editing

## Author ORCIDs
Timothy S Balmer  http://orcid.org/0000-0002-8864-5465

## Ethics
All animals were handled according to the approved institutional animal care and use committee (IACUC) protocol #21-1817R.

Reviewer #1 (Public Review): https://doi.org/10.7554/eLife.88321.4.sa1
Reviewer #2 (Public Review): https://doi.org/10.7554/eLife.88321.4.sa2
Author Response https://doi.org/10.7554/eLife.88321.4.sa3

# Additional files

## Supplementary files
• MDAR checklist

## Data availability
All data associated with this study are present in the article. The computational model is available at ModelDB: https://modeldb.science/2015953.

The following dataset was generated:

| Author(s) | Year | Dataset title | Dataset URL | Database and Identifier |
| --- | --- | --- | --- | --- |
| Balmer TS | 2023 | Unipolar brush cell circuits extend and diversify spiking patterns (Hariani et al., 2023) | https://modeldb.science/2015953 | ModelDB, 2015953 |

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
