## [Editor Report · eLife assessment]

This study presents **important** findings about synaptic connectivity among subsets of unipolar brush cells (UBCs), a specialized interneuron primarily located in the vestibular lobules of the cerebellar cortex. The evidence supporting the claims is interesting and **solid**. The work will be of interest to cerebellar neuroscientists as well as those focussed on synaptic properties and mechanisms. Although several **compelling** pieces of data were presented, some in vivo work remains to be conducted in order to test whether the hypothesis and predictions translate into the behaving animal and how it would impact the processing of feedback or feed-forward activity that would be required to promote behavior.

---

## [Referee Report · Reviewer #1 (Public Review)]

The manuscript by Hariani et al. presents experiments designed to improve our understanding of the connectivity and computational role of Unipolar Brush Cells (UBCs) within the cerebellar cortex, primarily lobes IX and X. The authors develop and cross several genetic lines of mice that express distinct fluorophores in subsets of UBCs, combined with immunocytochemistry that also distinguishes subtypes of UBCs, and they use confocal microscopy and electrophysiology to characterize the electrical and synaptic properties of subsets of so-labelled cells, and their synaptic connectivity within the cerebellar cortex. The authors then generate a computer model to test possible computational functions of such interconnected UBCs.

Using these approaches, the authors report that:

1. GRP-driven TDtomato is expressed exclusively in a subset (20%) of ON-UBCs, defined electrophysiologically (excited by mossy fiber afferent stimulation via activation of UBC AMPA and mGluR1 receptors) and immunocytochemically by their expression of mGluR1.

2. UBCs ID'd/tagged by mCitrine expression in Brainbow mouse line P079 is expressed in a similar minority subset of OFF-UBCs defined electrophysiologically (inhibited by mossy fiber afferent stimulation via activation of UBC mGluR2 receptors) and immunocytochemically by their expression of Calretinin. However, such mCitrine expression was also detected in some mGluR1 positive UBCs, which may not have shown up electrophysiologically because of the weaker fluorophore expression without antibody amplification.

3. Confocal analysis of crossed lines of mice (GRP X P079) stained with antibodies to mGluR1 and calretinin documented the existence of all possible permutations of interconnectivity between cells (ON-ON, ON-OFF, OFF-OFF, OFF-ON), but their overall abundance was low, and neither their absolute or relative abundance was quantified.

4. A computational model (NEURON ) indicated that the presence of an intermediary UBC (in a polysynaptic circuit from MF to UBC to UBC) could prolong bursts (MF-ON-ON), prolong pauses (MF-ON-OFF), cause a delayed burst (MF-OFF-OFF), cause a delayed pause (MF-OFF-ON) relative to solely MF to UBC synapses which would simply exhibit long bursts (MF-ON) or long pauses (MF-OFF).

The authors thus conclude that the pattern of interconnected UBCs provides an extended and more nuanced pattern of firing within the cerebellar cortex that could mediate longer lasting sensorimotor responses.

The cerebellum's long known role in motor skills and reflexes, and associated disorders, combined with our nascent understanding of its role in cognitive, emotional, and appetitive processing, makes understanding its circuitry and processing functions of broad interest to the neuroscience and biomedical community. The focus on UBCs, which are largely restricted to vestibular lobes of the cerebellum reduces the breadth of likely interest somewhat. The overall design of specific experiments is rigorous and the use of fluorophore expressing mouse lines is creative. The data that is presented and the writing are clear.

---

## [Referee Report · Reviewer #2 (Public Review)]

In this paper, the authors presented a compelling rationale for investigating the role of UBCs in prolonging and diversifying signals. Based on the two types of UBCs known as ON and OFF UBC subtypes, they have highlighted the existing gaps in understanding UBCs connectivity and the need to investigate whether UBCs target UBCs of the same subtype, different subtypes, or both. The importance of this knowledge is for understanding how sensory signals are extended and diversified in the granule cell layer.

The authors designed very interesting approaches to study UBCs connectivity by utilizing transgenic mice expressing GFP and RFP in UBCs, Brainbow approach, immunohistochemical and electrophysiological analysis, and computational models to understand how the feed-forward circuits of interconnected UBCs transform their inputs.

This study provided evidence for the existence of distinct ON and OFF UBC subtypes based on their electrophysiological properties, anatomical characteristics, and expression patterns of mGluR1 and calretinin in the cerebellum. The findings support the classification of GRP UBCs as ON UBCs and P079 UBCs as OFF UBCs and suggest the presence of synaptic connections between the ON and OFF UBC subtypes. In addition, they found that GRP and P079 UBCs form parallel and convergent pathways and have different membrane capacitance and excitability. Furthermore, they showed that UBCs of the same subtype provide input to one another and modify the input to granule cells, which could provide a circuit mechanism to diversify and extend the pattern of spiking produced by mossy fiber input. Accordingly, they suggested that these transformations could provide a circuit mechanism for maintaining a sensory representation of movement for seconds.

Overall, the article is well written in a sound detailed format, very interesting with excellent discovery and suggested model.

---

## [Author Response]

The following is the authors’ response to the previous reviews.

**eLife assessment**
This study presents valuable findings about synaptic connectivity among subsets of unipolar brush cells (UBCs), a specialized interneuron primarily located in the vestibular lobules of the cerebellar cortex. The evidence supporting the claims are interesting although incomplete in some areas. The work will be of interest to cerebellar neuroscientists as well as those focussed on synaptic properties and mechanisms. Although several compelling pieces of data were presented, substantial work remains to be conducted in order for the hypothesis and predictions of the manuscript to confirm how these factors play out in the actual brain circuit and how it would impact the processing of feedback or feedforward activity that would be required to promote behavior.
**Public Reviews:**

**Reviewer #1 (Public Review):**
The manuscript by Hariani et al. presents experiments designed to improve our understanding of the connectivity and computational role of Unipolar Brush Cells (UBCs) within the cerebellar cortex, primarily lobes IX and X. The authors develop and cross several genetic lines of mice that express distinct fluorophores in subsets of UBCs, combined with immunocytochemistry that also distinguishes subtypes of UBCs, and they use confocal microscopy and electrophysiology to characterize the electrical and synaptic properties of subsets of so-labelled cells, and their synaptic connectivity within the cerebellar cortex. The authors then generate a computer model to test possible computational functions of such interconnected UBCs.Using these approaches, the authors report that:1. GRP-driven TDtomato is expressed exclusively in a subset (20%) of ON-UBCs, defined electrophysiologically (excited by mossy fiber afferent stimulation via activation of UBC AMPA and mGluR1 receptors) and immunocytochemically by their expression of mGluR1.1. UBCs ID'd/tagged by mCitrine expression in Brainbow mouse line P079 is expressed in a similar minority subset of OFF-UBCs defined electrophysiologically (inhibited by mossy fiber afferent stimulation via activation of UBC mGluR2 receptors) and immunocytochemically by their expression of Calretinin. However, such mCitrine expression was also detected in some mGluR1 positive UBCs, which may not have shown up electrophysiologically because of the weaker fluorophore expression without antibody amplification.1. Confocal analysis of crossed lines of mice (GRP X P079) stained with antibodies to mGluR1 and calretinin documented the existence of all possible permutations of interconnectivity between cells (ON-ON, ON-OFF, OFF-OFF, OFF-ON), but their overall abundance was low, and neither their absolute or relative abundance was quantified.1. A computational model (NEURON ) indicated that the presence of an intermediary UBC (in a polysynaptic circuit from MF to UBC to UBC) could prolong bursts (MF-ON-ON), prolong pauses (MF-ON-OFF), cause a delayed burst (MF-OFF-OFF), cause a delayed pause (MF-OFF-ON) relative to solely MF to UBC synapses which would simply exhibit long bursts (MF-ON) or long pauses (MF-OFF).The authors thus conclude that the pattern of interconnected UBCs provides an extended and more nuanced pattern of firing within the cerebellar cortex that could mediate longer lasting sensorimotor responses.The cerebellum's long known role in motor skills and reflexes, and associated disorders, combined with our nascent understanding of its role in cognitive, emotional, and appetitive processing, makes understanding its circuitry and processing functions of broad interest to the neuroscience and biomedical community. The focus on UBCs, which are largely restricted to vestibular lobes of the cerebellum reduces the breadth of likely interest somewhat. The overall design of specific experiments is rigorous and the use of fluorophore expressing mouse lines is creative. The data that is presented and the writing are clear. However, despite some additional analysis in response to the initial review, the overall experimental design still has issues that reduce overall interpretation (please see specific issues for details), which combined with a lack of thorough analysis of the experimental outcomes undermines the value of the NEURON model results and the advance in our understanding of cerebellar processing in situ (again, please see specific issues for details).Specific issues:1. All data gathered with inhibition blocked. All of the UBC response data (Fig. 1) was gathered in the presence of GABAAR and Glycine R blockers. While such an approach is appropriate generally for isolating glutamatergic synaptic currents, and specifically for examining and characterizing monosynaptic responses to single stimuli, it becomes problematic in the context of assaying synaptic and action potential response durations for long lasting responses, and in particular for trains of stimuli, when feed-forward and feed-back inhibition modulates responses to afferent stimulation. I.e. even for single MF stimuli, given the >500ms duration of UBC synaptic currents, there is plenty of time for feedback inhibition from Golgi cells (or feedforward, from MF to Golgi cell excitation) to interrupt AP firing driven by the direct glutamatergic synaptic excitation. This issue is compounded further for all of the experiments examining trains of MF stimuli. Beyond the impact of feedback inhibition on the AP firing of any given UBC, it would also obviously reduce/alter/interrupt that UBC's synaptic drive of downstream UBCs. This issue fundamentally undermines our ability to interpret the simulation data of Vm and AP firing of both the modeled intermediate and downstream UBC, in terms of applying it to possible cerebellar cortical processing in situ.

The goal of Figure 1 was to determine the cell types of labeled UBCs in transgenic mouse lines, which is determined entirely by their synaptic responses to glutamate (Borges-Merjane and Trussell, 2015). Thus, blocking inhibition was essential to produce clear results in the characterization of GRP and P079 UBCs. While GABAergic/glycinergic feedforward and feedback inhibition is certainly important in the intact circuit, it was not our intention, nor was it possible, to study its contribution in the present study. Leaving inhibition unblocked does not lead to a physiologically realistic stimulation pattern in acute brain slices, because electrical stimulation produces synchronous excitation and inhibition by directly exciting Golgi cells, rather than their synaptic inputs. The main inhibition that UBCs receive that are crucial to determining burst or pause durations is not via GABA/glycine, but instead through mGluR2, which lasts for 100-1000s of milliseconds. The main excitation that drives UBC firing is mGluR1 and AMPA, which both last 100-1000s of milliseconds. Thus, these large conductances are unlikely to be significantly shaped by 1-10 ms IPSCs from feedforward and feedback GABA/glycine inhibition. Recent studies that examined the duration of bursting or pausing in UBCs had inhibition blocked in their experiments, presumably for the reasons outlined above (Guo et al., 2021; Huson et al., 2023).

Below is an example showing the synaptic currents and firing patterns in an ON UBC before and after blocking inhibition. The GABA/glycinergic inhibition is fast, occurs soon after the stimuli and has little to no effect on the slow inward current that develops after the end of stimulation, which is what drives firing for 100s of milliseconds.

**Author response image 1. sa3fig1:** Example showing small effect of GABAergic and glycinergic inhibition on excitatory currents and burst duration. (A) Excitatory postsynaptic currents in response to train of 10 presynaptic stimuli at 50 Hz before (black) and after (Grey) blocking GABA and glycine receptors. The slow inward current that occurs at the end of stimulation is little affected. (B) Expanded view of the synaptic currents evoked during the train of stimuli. GABA/glycine receptors mediate the fast outward currents that occur immediately after the first couple stimuli. (C) Three examples of the bursts caused by the 50 Hz stimulation in the same cell without blocking GABA and glycine receptors. (D) Three examples in the same cell after blocking GABA and glycine receptors.

The authors' response to the initial concern is (to paraphrase), "its not possible to do and its not important", neither of which are soundly justified.As stated in the original review, it is fully understandable and appropriate to use GABAAR/GlycineR antagonists to isolate glutamatergic currents, to characterize their conductance kinetics. That was not the issue raised. The issue raised was that then using only such information to generate a model of in situ behavior becomes problematic, given that feedback and lateral inhibition will sculpt action potential output, which of course will then fundamentally shape their synaptic drive of secondary UBCs, which will be further sculpted by their own inhibitory inputs. This issue undermines interpretation of the NEURON model.The argument that taking inhibition into account is not possible because of assumed or possible direct electrical excitation of Golgi cells is confusing for two interacting reasons. First, one can certainly stimulate the mossy fiber bundle to get afferent excitation of UBCs (and polysynaptic feedback/lateral inhibitory inputs) without directly stimulating the Golgi cells that innervate any recorded UBC. Yes, one might be stimulating some Golgi cells near the stimulating electrode, but one can position the stimulating electrode far enough down the white matter track (away from the recorded UBC), such that mossy fiber inputs to the recorded UBC can be stimulated without affecting Golgi cells near or synaptically connected to the recorded UBC. Moreover, if the argument were true, then presumably the stimulation protocol would be just as likely to directly stimulate neighboring UBCs, which then drove the recorded UBC's responses. Thus, it is both doable and should be ensured that stimulation of the white matter is distant enough to not be directly activating relevant, connected neurons within the granule cell layer.Finally, the authors present three examples of UBC recordings with and without inhibitory inputs blocked, and state "Thus, these large conductances are unlikely to be significantly shaped by 1-10 ms IPSCs from feedforward and feedback GABA/glycine inhibition" and "GABA/glycinergic inhibition...has little to no effect on the slow inward current that develops after the end of stimulation". This response reflects on original concerns about lack of quantification or consideration of important parameters. In particular, while the traces with and without inhibition are qualitatively similar, quantitative considerations indicate otherwise. First, unquantified examples are not adequate to drive conclusions. Regardless, the main issue (how inhibition affects actual responses in situ) is actually highlighted by the authors current clamp recordings of UBC responses, before and after blocking inhibition. The output response is dramatically different, both at early and late time points, when inhibition is blocked. Again, a lack of quantification (of adequate n's) makes it hard to know exactly how important, but quick "eye ball" estimates of impact include: (1) a switch from only low frequency APs initially (without inhibition blocked) to immediate burst of high frequency APs (high enough to not discern individual APs with given figure resolution) when inhibition is blocked, (2) Slow rising to a peak EPSP, followed by symmetrical return to baseline (without inhibition blocked) versus immediate rise to peak, followed by prolonged decay to baseline (with inhibition blocked), (3) substantially shorter duration (~34% shorter) secondary high frequency burst (individual APs not discernible) of APs (with inhibition blocked versus without inhibition blocked), and (4) substantial reduction in number of long delayed APs (with inhibition blocked versus without inhibition blocked). Thus, clearly, feedback/lateral inhibition is actually sculpting AP output at all phases of the UBC response to trains of afferent stimulations. Importantly, the single voltage clamp trace showing little impact of transient IPSCs on the slow EPSC do not take into account likely IPSC influences on voltage-activated conductances that would not occur in voltage-clamp recordings but would be free to manifest in current clamp, and thereby influence AP output, as observed.So again, our ability to understand how interconnected UBCs behave in the intact system is undermined by the lack of consideration and quantification of the impact of inhibition, and it not being incorporated into the model. At the very least a strong proviso about lack of inclusion of such information, given the authors' data showing its importance in the few examples shown, should be added to the discussion.

Thank you for this substantive explanation. Your points are well described and we agree that the single experiment shown is not strong evidence for a lack of importance of Golgi cell inhibition, especially on the temporal dynamics of spiking. Previous work has clearly shown that Golgi cells have several important roles in shaping the activity of the granular layer, including affecting the temporal dynamics of granule cell spikes. However, the work presented here focuses on the feedforward circuitry of UBCs and the large inward and large outward glutamatergic currents that drive spiking or pausing for 100s of milliseconds. Our model does not focus on the aspects that are most sensitive to Golgi cell inhibition, including timing of the first spikes in the UBC’s response. Nor does our model focus on short term plasticity, which we thought was reasonable because the slow currents in UBCs are quite insensitive to the temporal characteristics of glutamate release (See the example in the previous rebuttal). Our model does not include long term plasticity, which is also affected by Golgi cells. For these reasons we agree that the model presented does not explain how feedforward UBC circuits might “play out in the actual brain circuit and how it would impact the processing of feedback or feedforward activity that would be required to promote behavior.” We have included a new paragraph in the discussion clarifying the limitations of this study and the model, reproduced below.

"Limitations of the model

Here we addressed how feedforward glutamatergic excitation and inhibition is transformed from one UBC to the next depending on their subtype. The model focuses on AMPA receptor mediated excitation and mGluR2 mediated inhibition. One limitation of the model is that it does not consider feedforward and lateral inhibition from Golgi cells, which shape the spiking of UBCs in response to afferent stimulation. Golgi cells receive mossy fiber input and inhibit UBCs through their corelease of GABA and glycine (Dugue et al., 2005; Rousseau et al., 2012). Golgi cells control the temporal dynamics of the firing of granule cells as well as their gain (Rossi et al., 2003; Kanichay and Silver, 2008) and are critical to larger scale dynamics of the cerebellar cortical network (D‘Angelo, 2008). Purkinje cells provide additional inhibition to ON UBCs that could influence how UBC circuits transform signals (Guo et al., 2016). A more complex model that implements Golgi cells and other critical circuit elements will be needed to investigate the role of feedforward UBC circuits in cerebellar network dynamics and motor behaviors in vivo."

2. No consideration for involvement of polysynaptic UBCs driving UBC responses to MF stimulation in electrophysiology experiments. Given the established existence (in this manuscript and Dino et al. 2000 Neurosci, Dino et al. 2000 ProgBrainRes, Nunzi and Mugnaini 2000 JCompNeurol, Nunzi et al. 2001 JCompNeurol) of polysynaptic connections from MFs to UBCs to UBCs, the MF evoked UBC responses established in this manuscript, especially responses to trains of stimuli could be mediated by direct MF inputs, or to polysynaptic UBC inputs, or possibly both (to my awareness not established either way). Thus the response durations could already include extension of duration by polysynaptic inputs, and so would overestimate the duration of monosynaptic inputs, and thus polysynaptic amplification/modulation, observed in the NEURON model.

We are confident that the synaptic responses shown are monosynaptic for several reasons. UBCs receive a single mossy fiber input on their dendritic brush, and thus if our stimulation produces a reliable, short-latency response consistent with a monosynaptic input, then there is not likely to be a disynaptic input, because the main input is accounted for by the monosynaptic response. In all cells included in our data set, the fast AMPA receptor-mediated currents always occurred with short latency (1.24 ± 0.29 ms; mean ± SD; n = 13), high reliability (no failures to produce an EPSC in any of the 13 GRP UBCs in this data set), and low jitter (SD of latency; 0.074 ± 0.046 ms; mean ± SD; n = 13). These measurements have been added to the results section.

In some rare cases, we did observe disynaptic currents, which were easily distinguishable because a single electrical stimulation produced a burst of EPSCs at variable latencies. Please see example below. These cases of disynaptic input, which have been reported by others (Diño et al., 2000; Nunzi and Mugnaini, 2000; van Dorp and De Zeeuw, 2015) support the conclusion that UBCs receive input from other UBCs.

**Author response image 2. sa3fig2:** Example of GRP UBC with disynaptic input. Three examples of the effect of a single presynaptic stimulus (triangle) in a GRP UBC with presumed disynaptic input. Note the variable latency of the first evoked EPSC, bursts of EPSCs, and spontaneous EPSCs.

Author response: "UBCs receive a single mossy fiber input on their dendritic brush, and thus if our stimulation produces a reliable, short-latency response consistent with a monosynaptic input, then there is not likely to be a disynaptic input."This statement is not congruent with the literature, with early work by Mugnaini and colleagues (Mugnaini et al. 1994 Synapse; Mugnaini and Flores 1994 J. Comp. Neurol.) indicating that UBCs are innervated by 1-2 mossy fibers, which are as likely other UBC terminals as MFs. This leaves open the possibility that so called monosynaptic responses do, as originally suggested, already include polysynaptic feedforward amplification of duration. While the authors also indicate that isolated disynaptic currents can be observed when they occur in isolation, a careful examination and objective documentation of "monosynaptic" responses would address this issue. Presumably, if potential disynaptic UBC inputs occur during a monosynaptic MF response, it would be detected as an abrupt biphasic inward/outward current, due to additional AMPA receptor activation but further desensitization of those already active (as observed by Kinney et al. 1997 J. Neurophysiol: "The delivery of a second MF stimulus at the peak of the slow EPSC evoked a fast EPSC of reduced amplitude followed by an undershoot of the subsequent slow current"). If such polysynaptic inputs are truly absent and are "rare" in isolation, some estimation of how common or not such synaptic amplification is, would improve our understanding of the overall significance of these inputs.

We are confident that these currents are monosynaptic, because, as suggested, we carefully analyzed the latency, jitter and reliability, which was added to the previous revision. The latency and jitter are strong (quantitative) evidence that the first EPSC evoked was monosynaptic. While some UBCs have been reported to have multiple brushes, or brushes that branch and may contact multiple mossy fibers, or receive synaptic input onto their somas, these cases are rare in our experience in this age of mouse and there is no evidence for them in this dataset. For every trace we made a careful examination and documented that no delayed EPSCs were present. The presence of delayed EPSCs (or ‘abrupt biphasic inward/outward currents’ as described in Kinney et al 1997) would indeed suggest the presence of disynaptic activity or multiple inputs to the UBC, but these would be easily identified, even during a stimulation train. For these reasons we feel that we have established that polysynaptic feedforward amplification of duration is not present

We agree that the monosynaptic responses could be due to the stimulation of UBC axons. However, the absence of delayed EPSCs again suggests that if stimulation of a presynaptic UBC axon was producing the currents in the recorded UBC, then the axon was severed from the soma and AIS, because this region is necessary for the cell to produce more than a single spike per stimulation. We added a sentence describing the potential for the monosynaptic EPSCs to be due to the stimulation of presynaptic UBC axons.

Your point is well taken that a discussion of how common or rare these UBC to UBC connections is necessary to more clearly explain how we interpret their significance and we have expanded the paragraph in the discussion that does so. Thank you for this suggestion.

3. Lack of quantification of subtypes of UBC interconnectivity. Given that it is already established that UBCs synapse onto other UBCs (see refs above), the main potential advance of this manuscript in terms of connectivity is the establishment and quantification of ON-ON, ON-OFF, OFF-ON, and OFF-OFF subtypes of UBC interconnections. But, the authors only establish that each type exists, showing specific examples, but no quantification of the absolute or relative density was provided, and the authors' unquantified wording explicitly or implicitly states that they are not common. This lack of quantification and likely small number makes it difficult to know how important or what impact such synapses have on cerebellar processing, in the model and in situ.

As noted by the reviewer, the connections between UBCs were rare to observe. We decided against attempting to quantify the absolute or relative density of connections for several reasons. A major reason for rare observations of anatomical connections between UBCs is likely due to the sparse labeling. First, the GRP mouse line only labels 20% of ON UBCs and we are unable to test whether postsynaptic connectivity of GRP ON UBCs is the same as that of the rest of the population of ON UBCs that are not labeled in the GRP mouse line. Second, the Brainbow reporter mouse only labels a small population of Cre expressing cells for unknown reasons. Third, the Brainbow reporter expression was so low that antibody amplification was necessary, which then limited the labeled cells to those close to the surface of the brain slices, because of known antibody penetration difficulties. Therefore, we refrained from estimating the density of these connections, because each of these variables reduced the labeling to unknown degrees and we reasoned that extrapolating our rare observations to the total population would be inaccurate.

A paper that investigated UBC connectivity using organotypic slice cultures from P8 mice suggests that 2/3 of the UBC population receives UBC input, based on the observation that 2/3 of the mossy fibers did not degenerate as would be expected after 2 days in vitro if they were severed from a distant cell body (Nunzi and Mugnaini, 2000). It remains to be seen if this high proportion is due to the young age of these mice or is also the case in adult mice. Even if these connections are indeed rare, they are expected to have profound effects on the circuit, as each UBC has multiple mossy fiber terminals (Berthie and Axelrad, 1994), and mossy fiber terminals are estimated to contact 40 granule cells each (Jakab and Hamori, 1988). We have added a comment regarding this point to the discussion.

To address this issue, the authors added the following text to the discussion section: "We did not estimate the density of these UBC to UBC connections, because the sparseness of labeling using these approaches made an accurate calculation impossible. Previous work using organotypic slice cultures from P8 mice estimated that 2/3 of the UBC population receives input from other UBCs (Nunzi & Mugnaini, 2000), although it is unclear whether this is the case in older mice."While accurate, the addition doesn't really address the situation, which is that apparently the reported connections are rare. Adding the information about 2/3 of UBCs having UBC inputs in culture, implies the opposite might be true (i.e. that they might be quite common), which is in contrast to the authors' data, so should be reworded for clarity, which should also incorporate the considerations covered in point #2 above. I.e. if the authors do establish that none of their recordings have polysynaptic inputs, and if they determine that the number of cells that showed isolated di-synaptic inputs is indeed rare, then it suggests that these specific polysynaptic connections are in fact rare.

Thank you for pointing this out. We agree that adding this information is somewhat contradictory to our results and we have added more to this section in the discussion, provided below.

Anatomically identifiable connections between UBCs were not present in all brain slices and finding them required a careful search. UBC labeling was sparse due to the highly specific genetic labeling techniques and further sparsification by the Brainbow reporter, which made it impossible to estimate the density of these UBC to UBC connections. Electrophysiological evidences suggest that UBC to UBC connections are not common, because spontaneous EPSCs that would indicate a spontaneously firing presynaptic UBC are only rarely observed in UBCs recorded in acute brain slices. In an analysis of feedforward excitation of granule layer neurons, only 4 out of 140 UBCs had this indirect evidence of a firing presynaptic UBC (van Dorp and De Zeeuw, 2015), which suggests that UBC to UBC connections may be rare. On the other hand, previous work using organotypic slice cultures from P8 mice estimated that 2/3 of the UBC population receives input from other UBCs (Nunzi & Mugnaini, 2000). This suggests a much higher density of UBC to UBC connections, but could be due to the young age of the brains used, which is before UBCs have matured (Morin et al., 2001), and also due to increased collateral sprouting that can occur in culture (Jaeger et al., 1988). Another study imaged 2-4 week old rat cerebellar slices at an electron microscopic level and found that 4 out of 14 UBC axon terminals contacted UBC brushes (Diño et al., 2000). Future work is necessary to accurately estimate the density and impact of these feedforward UBC circuits.

4. Lack of critical parameters in NEURON model.A) The model uses # of molecules of glutamate released as the presumed quantal content, and this factor is constant.However, no consideration of changes in # of vesicles released from single versus trains of APs from MFs or UBCs is included. At most simple synapses, two sequential APs alters release probability, either up or down, and release probability changes dynamically with trains of APs. It is therefore reasonable to imagine UBC axon release probability is at least as complicated, and given the large surface area of contact between two UBCs, the number of vesicles released for any given AP is also likely more complex.B) the model does not include desensitization of AMPA receptors, which in the case of UBCs can paradoxically reduce response magnitude as vesicle release and consequent glutamate concentration in the cleft increases (Linney et al. 1997 JNeurophysiol, Lu et al. 2017 Neuron, Balmer et al. 2021 eLIFE), as would occur with trains of stimuli at MF to ON-UBCs.

A) The model produces synaptic AMPA and mGluR2 currents that reproduce those we recorded in vitro. We did not find it necessary to implement changes in glutamate release during a train as the model was fit to UBC data with the assumption that the glutamate transient did not change during the train. If there is a change in neurotransmitter release during a train, it is therefore built into the model, which has the advantage of reducing its complexity. UBCs are a special case where the postsynaptic currents are mediated mostly by the total amount of transmitter released. Most of the evoked current occurs tens to hundreds of milliseconds after neurotransmitter release and is therefore much more sensitive to total release and less sensitive to how it is released during the train. The figure below shows the effect of reducing the amount of glutamate released by 10% on each stimulus in the model. Despite a significant change in the pattern of neurotransmitter release, as well as a reduction in the total amount of glutamate, the slow EPSC still decays over the course of hundreds of milliseconds.

B) The detailed kinetic AMPA receptor model used here accurately reproduces desensitization, which in fact mediates that the slow ON UBC current. This AMPA receptor is a 13-state model, including 4 open states with 1-4 glutamates bound, 4 closed states with 1-4 glutamates bound, 4 desensitized states with 1-4 glutamates bound, and 5 closed states with 0-4 glutamates bound. The forward and reverse rates between different states in the model were fit to AMPA receptor currents recorded from dissociated UBCs and they accurately reproduced the ON UBC currents evoked by synaptic stimulation in our previous work (Balmer et al., 2021).

**Author response image 3. sa3fig3:** Effect of short-term depression of neurotransmitter release. (A) The top trace shows the glutamate transient that drives the AMPA receptor model used in our study. No change in release is implemented, although the slow tail of the transient summates during the train. The bottom trace shows the modeled AMPA receptor mediated current. (B) In this model the amount of glutamate released on each stimulus is reduced by 10%. The duration of the slow AMPA current is similar, despite a profound change in the pattern of neurotransmitter exposure.

While the authors have not added the suggested additional parameters, their clarifications regarding the implications of existing parameters, and demonstration of reasonable fits to experimental data, and lack of substantial effect of simulating reduced vesicle release probability,5. Lack of quantification of various electrophysiological responses. UBCs are defined (ON or OFF) based on inward or outward synaptic response, but no information is provided about the range of the key parameter of duration across cells, which seems most critical to the current considerations. There is a similar lack of quantification across cells of AP duration in response to stimulation or current injections, or during baseline. The latter lack is particularly problematic because in agreement with previous publications, the raw data in Fig. 1 shows ON UBCs as quiescent until MF stimulation and OFF UBCs firing spontaneously until MF stimulation, but, for example, at least one ON UBC in the NEURON model is firing spontaneously until synaptically activated by an OFF UBC (Fig. 11A), and an OFF UBC is silent until stimulated by a presynaptic OFF UBC (Fig. 11C). This may be expected/explainable theoretically, but then such cells should be observed in the raw data.

To address this reasonable concern of a general lack of quantification of electrophysiological responses we have added data characterizing the slow inward and outward currents evoked by synaptic stimulation in GRP and P079 UBCs in the results section and in new panels in Figure 1. We report the action potential pause lengths in P079 UBCs and burst lengths in ON UBCs in the results section. However, we favor the duration of the currents to the length of burst and pause, because the currents do not depend on a stable resting membrane potential, which is itself difficult to determine in intracellular recordings of these small cells. In a series of recent publications that focused on UBC firing, the authors argue that cell-attached recordings are necessary to determine accurately the burst and pause lengths, as well as spontaneous firing rates (Guo et al., 2021; Huson et al., 2023). (The trade-off of these extracellular recordings is that the monosynaptic nature of the input is nearly impossible to confirm.) Spontaneous firing rates were variable within both GRP and P079 UBCs from silent to firing regularly or in bursts, as previously reported (Kim et al., 2012; van Dorp and De Zeeuw, 2015). For clarity, we chose to model the GRP UBCs as silent unless receiving synaptic input and P079 UBCs as active unless receiving synaptic input. As the reviewer suggests, we have observed UBCs firing in the patterns similar to those shown in the model UBCs having input from spontaneous presynaptic UBCs. Below are some examples of spontaneous EPSCs and IPSCs in UBCs that suggest the presence of a presynaptic UBC.

**Author response image 4. sa3fig4:** Examples of UBCs that receive spontaneous input. (A) Three ON UBCs that had spontaneous EPSCs, suggesting the presence of an active presynaptic UBC. (B) Two OFF UBCs that had spontaneous outward currents.

The authors have added additional analysis and discussion, which adequately addresses this concern.
**Reviewer #2 (Public Review):**
In this paper, the authors presented a compelling rationale for investigating the role of UBCs in prolonging and diversifying signals. Based on the two types of UBCs known as ON and OFF UBC subtypes, they have highlighted the existing gaps in understanding UBCs connectivity and the need to investigate whether UBCs target UBCs of the same subtype, different subtypes, or both. The importance of this knowledge is for understanding how sensory signals are extended and diversified in the granule cell layer.The authors designed very interesting approaches to study UBCs connectivity by utilizing transgenic mice expressing GFP and RFP in UBCs, Brainbow approach, immunohistochemical and electrophysiological analysis, and computational models to understand how the feed-forward circuits of interconnected UBCs transform their inputs.This study provided evidence for the existence of distinct ON and OFF UBC subtypes based on their electrophysiological properties, anatomical characteristics, and expression patterns of mGluR1 and calretinin in the cerebellum. The findings support the classification of GRP UBCs as ON UBCs and P079 UBCs as OFF UBCs and suggest the presence of synaptic connections between the ON and OFF UBC subtypes. In addition, they found that GRP and P079 UBCs form parallel and convergent pathways and have different membrane capacitance and excitability. Furthermore, they showed that UBCs of the same subtype provide input to one another and modify the input to granule cells, which could provide a circuit mechanism to diversify and extend the pattern of spiking produced by mossy fiber input. Accordingly, they suggested that these transformations could provide a circuit mechanism for maintaining a sensory representation of movement for seconds.Overall, the article is well written in a sound detailed format, very interesting with excellent discovery and suggested model.I believe the authors have provided appropriate responses and have consequently revised the manuscript in a convincing manner. Although I am not an expert in physiology, I find the explanations and clarifications to be acceptable.